# Inhibition of type I PRMTs reforms muscle stem cell identity enhancing their therapeutic capacity

Claudia Dominici[1,2], Oscar D Villarreal[1], Junio Dort[3], Emilie Heckel[3], Yu Chang Wang[4], Ioannis Ragoussis[4], Jean-Sebastien Joyal[3], Nicolas Dumont[3], Stéphane Richard[1,2,5,6,7]*

[1]Segal Cancer Center, Lady Davis Institute for Medical Research, McGill University, Montreal, Canada; [2]Departments of Human Genetics, McGill University, Montreal, Canada; [3]CHU Sainte-Justine Research Center, Université de Montréal, Montréal, Canada; [4]Genome Quebec Innovation Centre, Montreal, Canada; [5]Gerald Bronfman, Department of Oncology, McGill University, Montréal, Canada; [6]Departments of Medicine, McGill University, Montreal, Canada; [7]Departments of Biochemistry, McGill University, Montréal, Canada

*For correspondence:
stephane.richard@mcgill.ca

Competing interest: The authors declare that no competing interests exist.

**Abstract** In skeletal muscle, muscle stem cells (MuSC) are the main cells responsible for regeneration upon injury. In diseased skeletal muscle, it would be therapeutically advantageous to replace defective MuSCs, or rejuvenate them with drugs to enhance their self-renewal and ensure long-term regenerative potential. One limitation of the replacement approach has been the inability to efficiently expand MuSCs ex vivo, while maintaining their stemness and engraftment abilities. Herein, we show that inhibition of type I protein arginine methyltransferases (PRMTs) with MS023 increases the proliferative capacity of ex vivo cultured MuSCs. Single cell RNA sequencing (scRNAseq) of ex vivo cultured MuSCs revealed the emergence of subpopulations in MS023-treated cells which are defined by elevated Pax7 expression and markers of MuSC quiescence, both features of enhanced self-renewal. Furthermore, the scRNAseq identified MS023-specific subpopulations to be metabolically altered with upregulated glycolysis and oxidative phosphorylation (OxPhos). Transplantation of MuSCs treated with MS023 had a better ability to repopulate the MuSC niche and contributed efficiently to muscle regeneration following injury. Interestingly, the preclinical mouse model of Duchenne muscular dystrophy had increased grip strength with MS023 treatment. Our findings show that inhibition of type I PRMTs increased the proliferation capabilities of MuSCs with altered cellular metabolism, while maintaining their stem-like properties such as self-renewal and engraftment potential.

## eLife assessment

This **valuable** paper informs on the role of type I PRMTs in programming muscle stem cell identification. The evidence presented is mostly **solid**, with some weaknesses in the evidence regarding the proposed mechanism. The paper will be of particular interest to those who study skeletal muscle satellite cell biology.

## Introduction

Muscle stem cells (MuSCs) reside on the periphery of muscle fibers in skeletal muscle and are activated following injury to drive muscle regeneration (*Lepper et al., 2011*; *Relaix and Zammit,*

*2012*). Major challenges arise when attempting to manipulate MuSCs ex vivo. Purified MuSCs spontaneously differentiate upon ex vivo expansion (*Fu et al., 2015a*; *Sacco et al., 2008*), limiting the potential of CRISPR Cas9-based applications and stem cell engraftment therapy. It is known that ex vivo cultured MuSCs are vastly inferior at engrafting and contributing to muscle regeneration compared to freshly isolated MuSCs, indicating the importance of retaining stemness for MuSC engraftment and muscle regeneration (*Montarras et al., 2005*; *Sakai et al., 2017*). Cell-intrinsic defects in patients with Duchenne muscular dystrophy (DMD) prevent MuSCs from maintaining a proper balance between stem cell self-renewal and differentiation, ultimately leading to depletion of regeneration-competent MuSCs (*Bentzinger et al., 2014*; *Dumont et al., 2015*; *Wang et al., 2013*).

MuSCs have metabolic flexibility that allows them to adapt to demands presented by their changing environments (*Relaix et al., 2021*). Indeed, metabolic signatures are strong contributors to the heterogeneity observed in isolated MuSCs (*Cho and Doles, 2017*). Several findings have indicated that the metabolic state of MuSCs is inextricably tied to their function and fate. Quiescent MuSCs predominantly rely on fatty acid oxidation (FAO) and oxidative phosphorylation (OxPhos; *Machado et al., 2017*; *Ryall et al., 2015*). While glycolysis remains low in quiescent cells, it is upregulated as MuSCs begin to proliferate following activation, representing a metabolic shift in activated MuSCs (*L'honoré et al., 2018*; *Ryall et al., 2015*). The dynamic regulation of metabolic state during the transition from MuSC quiescence to proliferation therefore presents a unique avenue to regulate MuSC regenerative capacity.

There is growing evidence to suggest a role for the post-translational modification arginine methylation in muscle cellular metabolism (*Blanc and Richard, 2017*; *Saber and Rudnicki, 2022*; *vanLieshout and Ljubicic, 2019*). Arginine methylation is carried out by 9 protein arginine methyltransferases (PRMTs; *Xu and Richard, 2021*). PRMTs are classified according to their catalytic activity: type I enzymes (ex: PRMT1, PRMT2, PRMT3, CARM1, PRMT6 and PRMT8) catalyze arginine asymmetrical dimethylation (ADMA), type II enzymes (ex: PRMT5, PRMT9) catalyze arginine symmetrical dimethylation (SDMA), and the unique type III enzyme PRMT7 catalyzes arginine monomethylation (MMA) (*Bedford and Clarke, 2009*; *Jain and Clarke, 2019*). PRMTs methylate many cellular proteins including histones and are known epigenetic regulators (*Xu and Richard, 2021*).

PRMTs are known regulators of MuSCs. PRMT1-null mouse MuSCs are highly proliferative and can be expanded in culture, but are unable to terminate differentiation, leading to a severe muscle regeneration defect in vivo following muscle injury (*Blanc et al., 2017*). CARM1 is a well-known transcriptional coactivator for nuclear receptors and p300/CPB (*Bedford and Clarke, 2009*) and it is a mediator of skeletal muscle plasticity (*Saber and Rudnicki, 2022*; *vanLieshout and Ljubicic, 2019*). CARM1 methylates the myogenic transcription factor, PAX7, to complex with histone methyltransferase complex to allow the expression of myogenic transcription factor Myf5 during asymmetric stem cell divisions (*Kawabe et al., 2012*). CARM1 regulates MuSCs metabolism by methylating AMP-activated protein kinase (AMPK) in MuSCs (*Stouth et al., 2020*). PRMT5 regulates MuSC proliferation by epigenetically downregulating expression of the cell cycle inhibitor p21 and deletion of PRMT5 in MuSCs blocks their expansion ex vivo and causes defects in muscle regeneration in vivo (*Zhang et al., 2015*). PRMT7 functions in the myogenic process and is required for MuSC self-renewal and muscle regeneration, as deletion of PRMT7 in MuSCs leads to persistent p21 expression (*Blanc et al., 2016*).

Given the importance in maintaining MuSC stemness and the enhanced MuSC proliferation observed with PRMT1 deficiency (*Blanc et al., 2017*), we aimed to determine whether the reversible inhibition of type I PRMTs using the MS023 inhibitor (*Eram et al., 2016*) would affect cellular metabolism and fulfil the requirements of generating a stem-like MuSC in culture with subsequent regenerative capabilities. Small molecule inhibitors of epigenetic regulators are a promising avenue to pursue in the treatment of muscle wasting diseases. Indeed, an inhibitor of Setd7-mediated lysine methylation (*Judson et al., 2018*), was shown to have beneficial effects on self-renewal and regeneration.

Herein, we show that MS023, an inhibitor of type I PRMTs, fulfiled the requirements of a reversible inhibitor which maintained the stemness of cultured MuSCs, while allowing for engraftment and regenerative capabilities. Although type I PRMTs have well-known indications as epigenetic modulators, we show that targeting type I PRMTs promoted the proliferation of MuSCs and maintained their self-renewal and ability to engraft.

## Results

### Type I PRMT inhibitor MS023 enhances self-renewal and in vitro expansion of MuSCs

We isolated single muscle fibers from 6- to 8-week-old C57BL/6 mice, cultured them in vitro with MS023 or vehicle (DMSO) for 48 hr, and stained them for the myogenic transcription factor Pax7 (*Seale et al., 2000*; *Soleimani et al., 2012*), and the proliferation marker ki67. The number of Pax7+/ki67+ cells increased significantly in the presence of MS023 (*Eram et al., 2016*) for 48 hr (*Figure 1A and B*). We observed a significant increase in the number of clusters with 3 or more cells with MS023 treatment (*Figure 1B*), as observed for PRMT1-deficient MuSCs (*Blanc et al., 2017*). Treatment with a specific CARM1 inhibitor TP-064 (*Nakayama et al., 2018*) over the course of 6 days did not increase MuSC cell number, ruling out that MS023 increased proliferation was mediated by CARM1 (*Figure 1—figure supplement 1A*). Moreover, we investigated the expression of other type I PRMTs from publicly available single cell RNAseq datasets (*Oprescu et al., 2020*), which were performed on skeletal muscle at different time points post-cardiotoxin injury (uninjured, and 0.5, 2, 3.5, 5, 10, and 21 days post-injury). Their findings show that PRMT1 is by far the most expressed type I PRMT in MuSCs at every time point tested (*Figure 1—figure supplement 1B*). CARM1 (PRMT4) is expressed at high level in a small/moderate subset of cells, especially during regeneration. PRMT6 is expressed at low level in a small proportion of cells, while PRMT8 expression was not detected. These findings show that PRMT1 is the predominant type I PRMT1 in MuSCs. Taken together, these data show that type I PRMT inhibition causes in vitro MuSC expansion and is likely mediated by the inhibition of PRMT1.

To determine the activation state of these fiber-associated MuSCs, we examined fibers after 48 hr in culture for the expression of Pax7 and the myogenic regulator transcription factor, MyoD (*Rudnicki et al., 1993*; *Troy et al., 2012*). Interestingly, MS023 treatment restricted the proportion of committed Pax7-/MyoD+ cells, while expanding the proportion of activated and cycling cells (Pax7+/MyoD+, *Figure 1C and D*).

MuSCs spontaneously differentiate into non-cycling myoblasts and lose their stemness in vitro (*Montarras et al., 2005*; *Ryall et al., 2015*; *Sacco et al., 2008*). The enhanced proliferation of MuSCs observed with type I PRMT inhibition on muscle fibers prompted us to examine whether similar effects could be achieved with FACS-purified MuSCs cultured in vitro. MuSCs incubated with MS023 for 48 hr had reduced ADMA containing proteins, as visualized by immunoblotting (*Figure 1—figure supplement 1C*), and had increased proliferative capacity of MuSCs, as determined by ki67 expression by immunofluorescence and FACS analysis (*Figure 1—figure supplement 1D and E*). These data are consistent with what was observed in muscle fiber-associated MuSCs (*Figure 1A–C*). Importantly, immunofluorescence analysis of myogenic markers in cultured MuSCs revealed that MS023 treatment prevented initiation of differentiation, as indicated by the significantly reduced sub-population of committed Pax7-/MyoD+ cells compared to control DMSO culture (13% vs 31%, respectively) (*Figure 1E and F*). Additionally, there was an increase in the proportion of cells that have retained their stemness by delaying expression of MyoD (Pax7+/MyoD-) in the MS023-treated samples compared to DMSO (40% vs 9%, *Figure 1E and F*). Similar proportions were observed with fiber-associated MuSCs (34% vs 14%, *Figure 1C and D*). Furthermore, we observed that the impediment of precocious differentiation observed by treatment with MS023 led to enhanced long-term culture capabilities of MuSCs up to the 14th passage (*Figure 1—figure supplement 1F*). Altogether, these findings indicate that treatment with MS023 maintained cultured MuSCs in a stem-like state.

### Transcriptionally distinct MuSC sub-populations emerge under type I PRMT inhibition

To identify subpopulations of MuSCs generated by MS023, we performed single-cell RNA sequencing (scRNAseq). MuSCs were purified from 8-week-old C57BL/6 mice in biological duplicates immediately after (1) isolation (termed d0; sample day 0), and (2) culture in growth medium for 4 days with 0.033% DMSO as control (sample d4) or with 1 µM MS023 (sample d4MS023), and (3) grown in growth media for 6 days with 0.033% DMSO removed at day 4 (sample d6), or 6 days in culture with 1 µM MS023 removed at day 4 (sample d6MS023rd4). Cells were collected at these time points and processed on the 10 x Genomics Chromium platform for scRNAseq. Data filtering and analysis was performed using

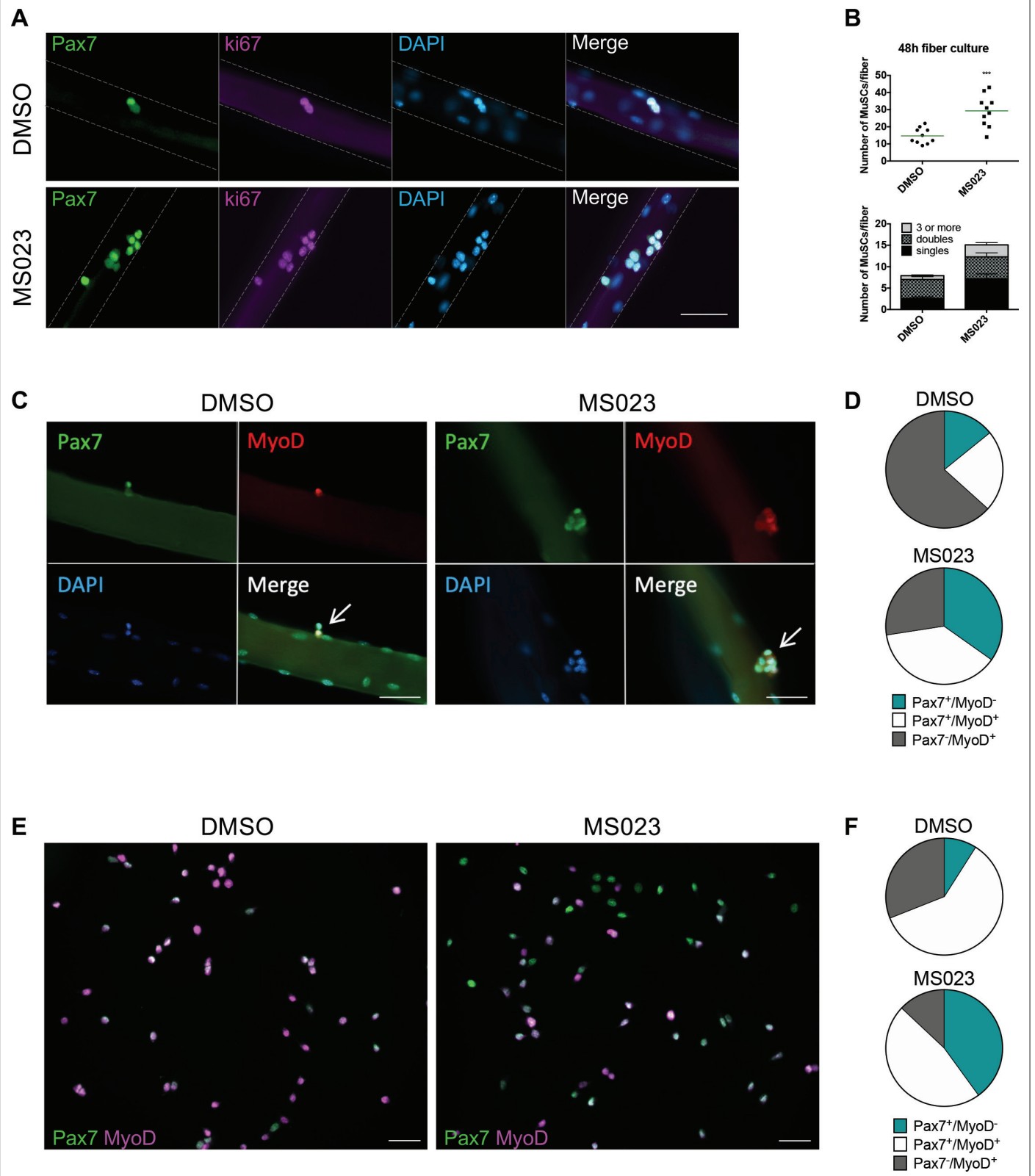

**Figure 1.** Enhanced self-renewal of MS023-treated MuSCs. (**A**) Muscle fibers cultured for 48 hr with MS023 or DMSO. Pax7 identifies MuSCs, and ki67 identifies proliferating cells. Scale bar represents 50 μM. (**B**) Quantification of total MuSCs per fiber. Thirty fibers were counted per condition, horizontal bar represents average number of MuSCs per fiber (n=3 mice per condition). (**C**) Muscle fibers cultured for 48 hr with MS023 or DMSO. 30 fibers counted per condition (n=3 mice per condition). Scale bar represents 50 μM. (**D**) Quantification of Pax7/MyoD expressing MuSCs from (**C**). (**E**) Pax7/MyoD

*Figure 1 continued on next page*

Figure 1 continued

immunofluorescence staining of cultured myoblasts treated with MS023 or DMSO for 48 hr. Scale bar represents 50 μM. (**F**) Quantification of Pax7/MyoD expressing cells from (**E**).

The online version of this article includes the following source data and figure supplement(s) for figure 1:

**Figure supplement 1.** Type I PRMT inhibitor MS023 increases proliferation of MuSCs most by inhibition of PRMT1.

**Figure supplement 1—source data 1.** Uncropped immunoblot.

the Seurat package for scRNAseq analysis (*Butler et al., 2018*). After filtering out low-quality cells, approximately 4000 cells (average, 3,887±44 cells) from each sample were retained for subsequent analysis (*Supplementary file 2*). A Pearson correlation of >0.98 (*Figure 2—figure supplement 1A*) was obtained between replicates and we chose to include a replicate for each time point for analysis.

Dimensionality reduction was performed using uniform manifold approximation and projection (UMAP) on pooled cells from all samples, resulting in 10 distinct subpopulations of myogenic cells (labelled 0–9, *Figure 2—figure supplement 1B*). We then visualized the distribution of individual samples within the UMAP embedding (*Figure 2C*, *Figure 2—figure supplement 1C*). To determine the identity of the 10 subpopulations, we examined the expression of *Pax7* and the bHLH transcription factors *Myod1*, *Myf5,* and *Myog* (*Figure 2B*).

Two populations of *Pax7* and *Myf5* expressing cells were observed that we called quiescence 1 and 2 (Q1 and Q2), while the two populations that expressed both *Myod1* and *Myog* were called differentiated myoblasts 1 and 2 (DM1 and DM2, *Figure 2A*). The remaining *Myod1* populations we named myoblast clusters 1–5 (M1-M5), and the last one C1 for committed 1 (*Figure 2A*). The freshly isolated MuSCs of day 0 (d0) segregated into Q1 and Q2 (*Figure 2—figure supplement 1D*, *Supplementary file 3*), which in addition to *Pax7* and *Myf5*, expressed markers of MuSC quiescence (*Fos, Egr1, Jun*), and early activation markers (*Myod1, Mt1, Mt2, Supplementary file 1*). Our Q1 and Q2 populations are similar to quiescence and early activation populations reported recently from freshly isolated (*De Micheli et al., 2020*; *Dell'Orso et al., 2019*; *van Velthoven et al., 2017*). The two *Myog* expressing clusters expressed other markers of early differentiation (*Acta2*; DM1), and late differentiation (*Tnnt3, Myl1, Acta1*; DM2) (*Supplementary file 1*). An enrichment for cell cycle and proliferation markers was found throughout the remaining 5 undifferentiated myoblast clusters (*Cdk1, Ccnd1, Mki67, Top2a, Birc5,* and *Cenpa*; M1-M5) (*Supplementary file 1*).

The control DMSO day 4 (d4) sample was distributed among three subpopulations of myoblasts (M1, M2, and M4), while two additional days in culture day 6 (d6) contained M1, M2, and M4, but also had differentiated clusters DM1 and DM2 (*Figure 2C and D*; *Figure 2—figure supplement 1E*, *Supplementary file 3*). In comparison, day 4 with MS023 (d4MS023) was distributed uniquely among clusters M3 and M5. Day 6 MS023 removed at day 4 (d6MS023rd4) had clusters of day 4 MS023 treatment, as well as DM1 and DM2 (*Figure 2C and D*, *Figure 2—figure supplement 2A and B*, *Supplementary file 3*). Notably, d6MS023rd4 had nearly twice as many cells in DM1 (20.8%) as d6 (11.5%; *Figure 2D*), indicating that the differentiation block induced by MS023 was reversible. In addition, all samples had in common a small subpopulation of cells (<200 cells) in cluster C1 (*Supplementary file 3*).

Analysis of the top 100 enriched genes in each cluster revealed distinct transcriptional profiles (*Supplementary file 1*). For example, the d4MS023-occupied cluster M5 was enriched for cell cycle markers (*Cdk1, Cdk4, Ccnd3*) and other markers of proliferation (*Mki67, Birc5, Top2a*) (*Supplementary file 1*). The other d4MS023-occupied cluster M3 retained some markers of proliferation (*Ccnd1*), however also expressed the cell cycle inhibitor *Cdkn2a* and the structural muscle gene *Myl6*, in addition to a strong enrichment for ribosomal genes (over 20% of the top 100 markers belonged to the ribosomal protein family, a feature that is unique to cluster M3). Therefore, clusters M3 and M5 are unique to MS023-treated MuSCs and represent proliferation clusters with extensive ribosomal protein expression.

The top 100 enriched genes in each sample were also investigated (*Supplementary file 2*, Heatmap in *Figure 2—figure supplement 3*). Interestingly, d4MS023 genes were strongly enriched for the Reactome Pathway of respiratory electron transport, the citric acid cycle, and formation of ATP. In contrast, d4 was enriched for the Reactome Pathways of RNA metabolism, translation, and mRNA splicing (*Figure 2—figure supplement 2C and D*).

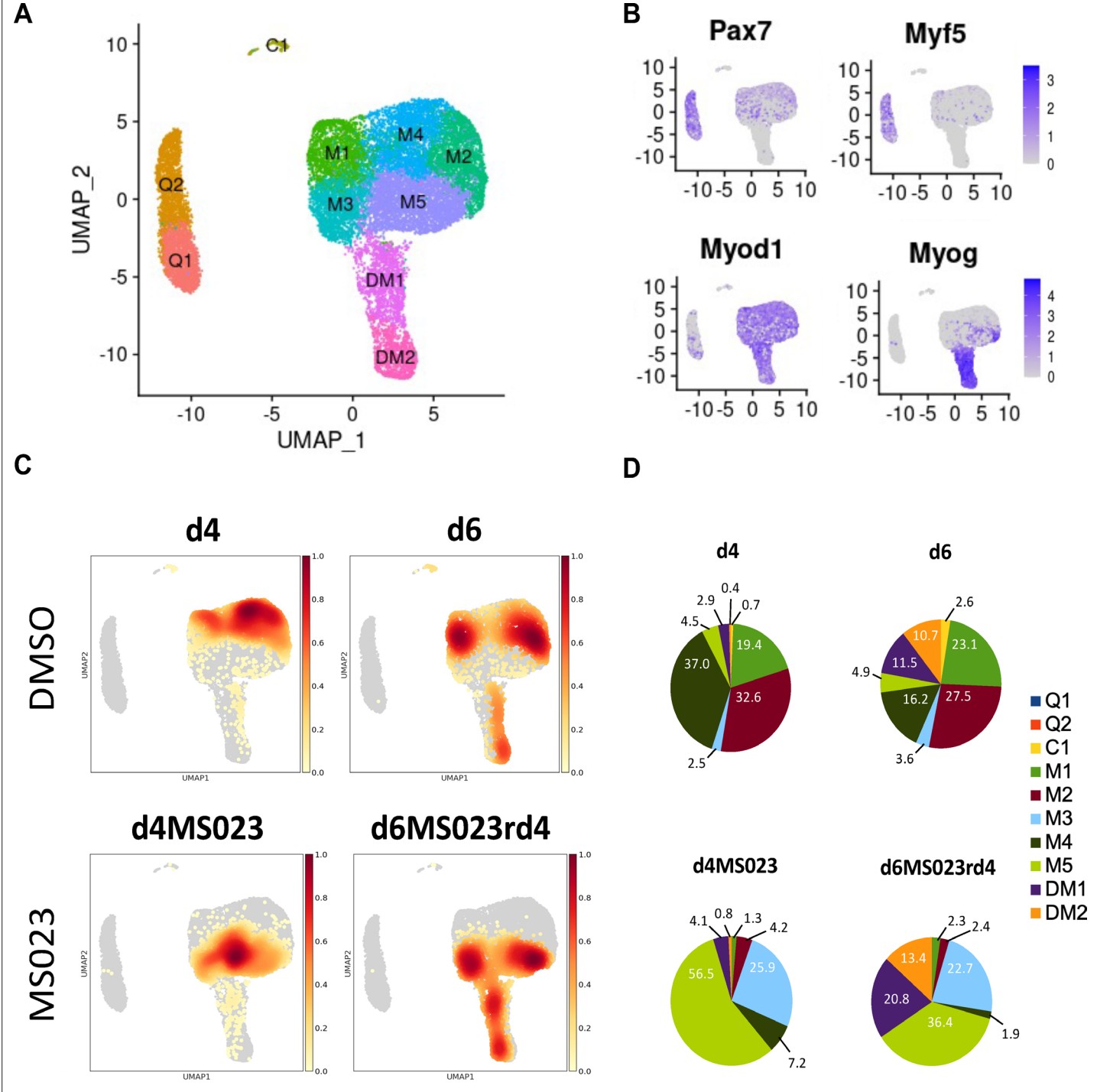

**Figure 2.** Single-cell graph-based clustering analysis. (**A**) UMAP embedding representation for all cells. (**B**) Gene expression density on the UMAP embedding plot of myogenic markers Pax7, Myf5, MyoD, and Myogenin. (**C**) Distribution of cells from each sample within the UMAP embedding representation. (**D**) Proportion of cells from each sample belonging to each of the 10 identified clusters.

The online version of this article includes the following figure supplement(s) for figure 2:

**Figure supplement 1.** scRNAseq analysis of DMSO- and MS023-treated MuSCs.

**Figure supplement 2.** scRNAseq analysis of DMSO- and MS023-treated MuSCs.

**Figure supplement 3.** scRNAseq analysis of DMSO- and MS023-treated MuSCs.

The d4 cluster M2 shared many of the same gene enrichments as cluster M5 (*Cdk1*, *Mki67*, *Birc5*, *Top2a*) (*Supplementary file 1*). Conversely, the other d4 cluster M4 down-regulated these genes while maintaining expression of replication-dependent transcripts such as *Ube2c* and *Dut*, indicating increased heterogeneity in cycling cells compared to d4MS023 (*Supplementary file 1*). The final d4 cluster M1 resembled the transitional d4MS023 cluster M3 in that it expressed a mix of cell cycle (*Ccnd1*) and differentiation (*Myl6*) markers, however lacked the enrichment of ribosomal genes observed in M3 (*Supplementary file 1*).

It was important to note is the stark difference in the pathway enrichment signatures of the transitional clusters M1 (d4) and M3 (d4MS023). The genes which were uniquely enriched in cluster M3 were strongly associated with protein synthesis (e.g., *Rps9*, *Eef1g*, *Uba52*) (*Supplementary file 1*, *Figure 2—figure supplement 2C and D*). In contrast, the enriched genes unique to cluster M1 were associated with cellular response to toxic substance and cellular detoxification (*Anxa1*, *Txn1*, *Hmox1*, *Prdx5*) (*Figure 2—figure supplement 2C and D*).

We further investigated the distribution of expression levels for *Pax7*, *Myf5*, *Myod1*, and *Myog* across clusters and samples (*Figure 2—figure supplement 2A and B*). We observed a unique expression pattern for *Pax7* in d4MS023 and d6MS023rd4 samples (*Figure 2—figure supplement 2B*). In d4 MuSCs, cells were divided into two populations, *Pax7* negative (expression level 0) and *Pax7* positive (standard distribution around expression level 1). In d6 myoblasts, *Pax7* positive cells maintain the same level of *Pax7* expression, but a higher proportion of cells became *Pax7* negative, consistent with canonical *Pax7* expression during MuSC differentiation (*Yin et al., 2013*). Meanwhile, with MS023 treatment, three subpopulations emerged, a *Pax7* negative (expression level 0) and two *Pax7*-positive cell (standard distribution around expression level 1) populations. The *Pax7*-positive cell populations in d4MS023 and d4MS023rd4 had higher *Pax7* expression levels than their untreated counterparts and were more comparable with day 0 (quiescent) MuSC *Pax7* expression levels (*Figure 2—figure supplement 2B*).

## Trajectory analysis reveals temporally distinct expression patterns in MuSCs treated with MS023 with an enrichment for energy metabolism genes

Using the Monocle v2.16.0 R package (*Trapnell et al., 2014*), we conducted trajectory analysis on 3 groups of samples to obtain hierarchical links between clusters: (1) all samples; (2) d0, d4, d6; (3) d0, d4MS023, d6MS023rd4. Monocle analysis placed each cell from the 9 clusters (Q1/Q2, M1-5, and DM1/2) onto a pseudotime axis based on differential gene expression patterns. The trajectory analysis of all samples pooled produced a pseudotime axis with a single branchpoint and 3-cell states which we labelled Q for quiescence, M for myoblast, and DM for differentiating myoblast (*Figure 3A*). Cells from the Q1 and Q2 clusters were found at the beginning of pseudotime, cells from the M1-5 clusters populated the area around the branchpoint, and cells from DM1/2 mainly populated the upward-oriented branch, following an overall trajectory of Q1/2→M1-5→DM1/2 (*Figure 3—figure supplement 1A*).

Day 0 cells populated the beginning of pseudotime, with d4 and d4MS023 cells appearing shortly after. d6 and d6MS023rd4 cells had a similar distribution as d4 and d4MS023, with the exception of having increased density around the terminus of the upward-oriented branch (*Figure 3—figure supplement 1A*). Interestingly, d4MS023 and d6MS023rd4 cells emerge later along the pseudotime axis than d4 and d6 cells and appeared closer to the branchpoint (*Figure 3—figure supplement 1A*).

We examined the genes which were significantly differentially regulated across the branchpoint of the pooled sample trajectory to confirm whether this split is representative of the molecular events which determine whether MuSCs will commit to differentiation or continue to proliferate. As expected, genes involved in the myogenic process were highly upregulated along the branch that diverges upwards along the trajectory (e.g. *Tnnt1*, *Tnnt2*, *Acta1*, *Myog*), while genes involved in cell cycle and proliferation were upregulated along the branch that diverges downwards along the trajectory (e.g. *Mki67*, *Cdca8*, *Top2a*, *Birc5*) (*Figure 3—figure supplement 1B and C*).

Trajectory analysis was also performed on a subset of samples (d0→d4→d6; DMSO trajectory and d0→d4MS023→d4MS023rd4; MS023 trajectory) to identify treatment-specific alterations in the molecular events which drive fate-determining decisions (*Figure 3B, C, E and F*). Both the DMSO and

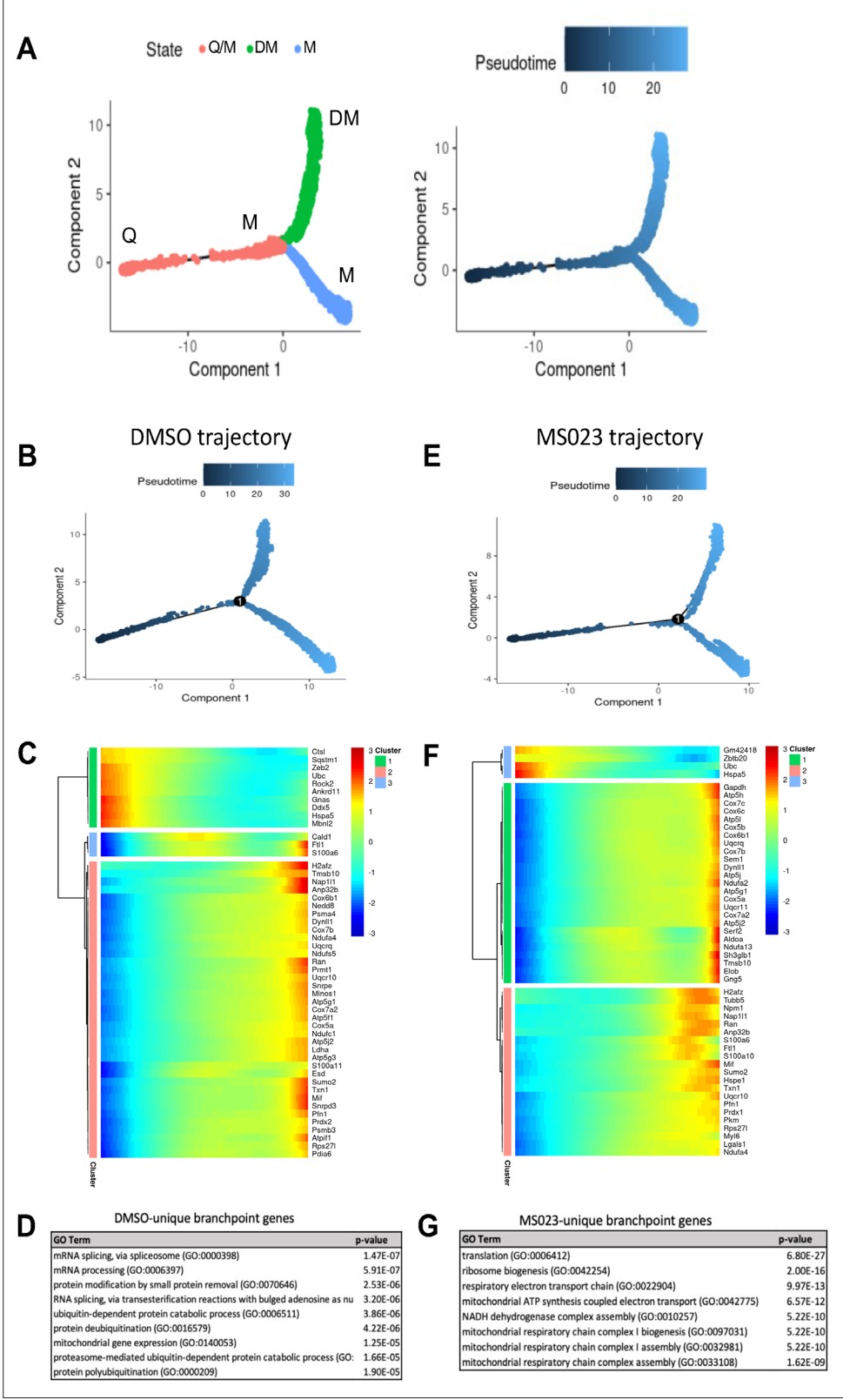

**Figure 3.** Pseudotime trajectory of pooled DMSO- and MS023-treated MuSCs. (**A**) Trajectory of all cells depicting three cell states. Pseudotime starts at the left endpoint of the plot. Q: Quiescence. M: Proliferating myoblast. DM: Differentiating myoblast. (**B–D**) Monocle trajectory of day-0 and DMSO-treated samples (**B**), heatmap of top differentially regulated genes across pseudotime (**C**) and GO enrichment analyses of DMSO-unique branchpoint

*Figure 3 continued on next page*

*Figure 3 continued*

genes (**D**). (**E–G**) Monocle trajectory of day-0 and MS023-treated samples (**E**), heatmap of top differentially regulated genes across pseudotime (**F**) and GO enrichment analyses of MS023-unique branchpoint genes (**G**).

The online version of this article includes the following figure supplement(s) for figure 3:

**Figure supplement 1.** Monocle trajectory analysis of DMSO- and MS023-treated MuSCs.

MS023 trajectories retained a single branch point indicating that the overall lineage patterns were conserved.

Closer inspection of genes which were significantly differentially regulated across the branch-point in the DMSO and MS023 trajectories revealed a unique signature in the MS023-trajectory. The list of genes which changed significantly across the branchpoint for both trajectories (7094 for the DMSO trajectory, 7616 for the MS023 trajectory) were filtered for genes which were unique to each condition. The DMSO trajectory had 1517 unique genes, while the MS023 trajectory had 2037 unique genes (*Figure 3—figure supplement 1D*). Interestingly, the MS023 trajectory-unique genes revealed an enrichment for oxidative phosphorylation and mitochondrial ATP synthesis (*Figure 3F and G*), consistent with cluster marker analysis for MS023-specific clusters ((*Figure 2—figure supplement 2C and D*): *electron transport, ATP synthesis*). Expression of select genes involved in the mitochondrial electron transport chain (*Ndua2, Ndufa4, Uqcr10, Cox4l1, Cox5b, Cox8a, Atp5c1, AtpIf1, Atp5j2*) were further examined for expression level within the scRNAseq UMAP plot, and it was confirmed that they were elevated in MS023-specific clusters (*Figure 4A*). More-over, RT-qPCR analysis of a subset of these genes confirmed they were upregulated in d4MS023 MuSCs compared to d4 MuSCs (*Figure 4—figure supplement 1A*). Therefore, these pathways both define the identity of MS023-treated MuSCs, and these pathways may serve a role in fate determination.

The transition of MuSC from quiescence to proliferation is accompanied by a sharp reduction in oxidative metabolism and a shift towards glycolysis (*L'honoré et al., 2018*; *Ryall et al., 2015*). Therefore, the retention of a transcriptional signature indicating high levels of mitochondrial oxidative phosphorylation in MS023-treated cells was unexpected. We sought to validate these findings in ex vivo cultured MuSCs by performing metabolic analysis with Seahorse Extracellular Flux Analyzer. We isolated MuSCs from C57BL/6 mice and seeded them onto matrigel-coated 96-well plates treated with 1 µM MS023 or DMSO for 48 hr (n=3 mice per treatment condition). Oxygen consumption rate (OCR) was first measured to characterize mitochondrial bioenergetics (*Figure 4B*). As predicted, MS023-treated MuSCs displayed increased OCR measurements at nearly all timepoints (*Figure 4B*, p=0.000001–0.02), with significantly elevated basal respiration (p=0.006) and maximal respiration (p=0.04), and a trend of increased ATP production (approaching statistical significance, p=0.1). To confirm an increase in mitochondrial biogenesis, d4 and d4MS034 MuSCs were stained with MitoTracker CMXRos (Thermo Fisher Scientific). The average intensity of the MitoTracker signal was significantly increased in d4MS023 cells (*Figure 4C and D*, p<0.0001).

We next determined whether markers of glycolysis were also differentially regulated in MS023-treated MuSCs. Indeed, several of the marker genes for MS023-specific clusters were components of glycolysis which displayed increased expression compared to clusters of DMSO-treated cells (*Hk1, Eno1, Pfkm, Gapdh, Pkm, Slc2a1*; *Figure 4—figure supplement 1B*). We next performed extracellular acidification rate (ECAR) analysis as an indicator of glycolysis again using the Seahorse Extracellular Flux Analyzer. MS023-treated cells were strikingly more glycolytic than DMSO-treated cells (*Figure 4—figure supplement 1C*). Glycolysis, glycolytic capacity, and glycolytic reserve were all significantly increased (p=0.02, p=0.004 and p=0.001, respectively) (*Figure 4—figure supplement 1C*). As further indicators of glycolysis, the growth media supernatant of d4 and d4MS023 cultured cells was analyzed using a Nova Biomedical Bioprofile 400 analyzer. As expected, d4MS023 cells had significantly higher levels of glucose uptake (measured as grams/liter per 10,000 cells, p=0.004). Additionally, the glycolytic byproduct lactate was present in higher concentrations d4MS023 media and was undetectable in the d4 media (*Figure 4—figure supplement 1D*). Collectively, these findings indicate that MS023-treated MuSCs utilize both OxPhos and glycolysis at elevated levels compared to untreated control cells to boost cellular metabolism, suggesting a possible mechanism through which enhanced proliferation is supported.

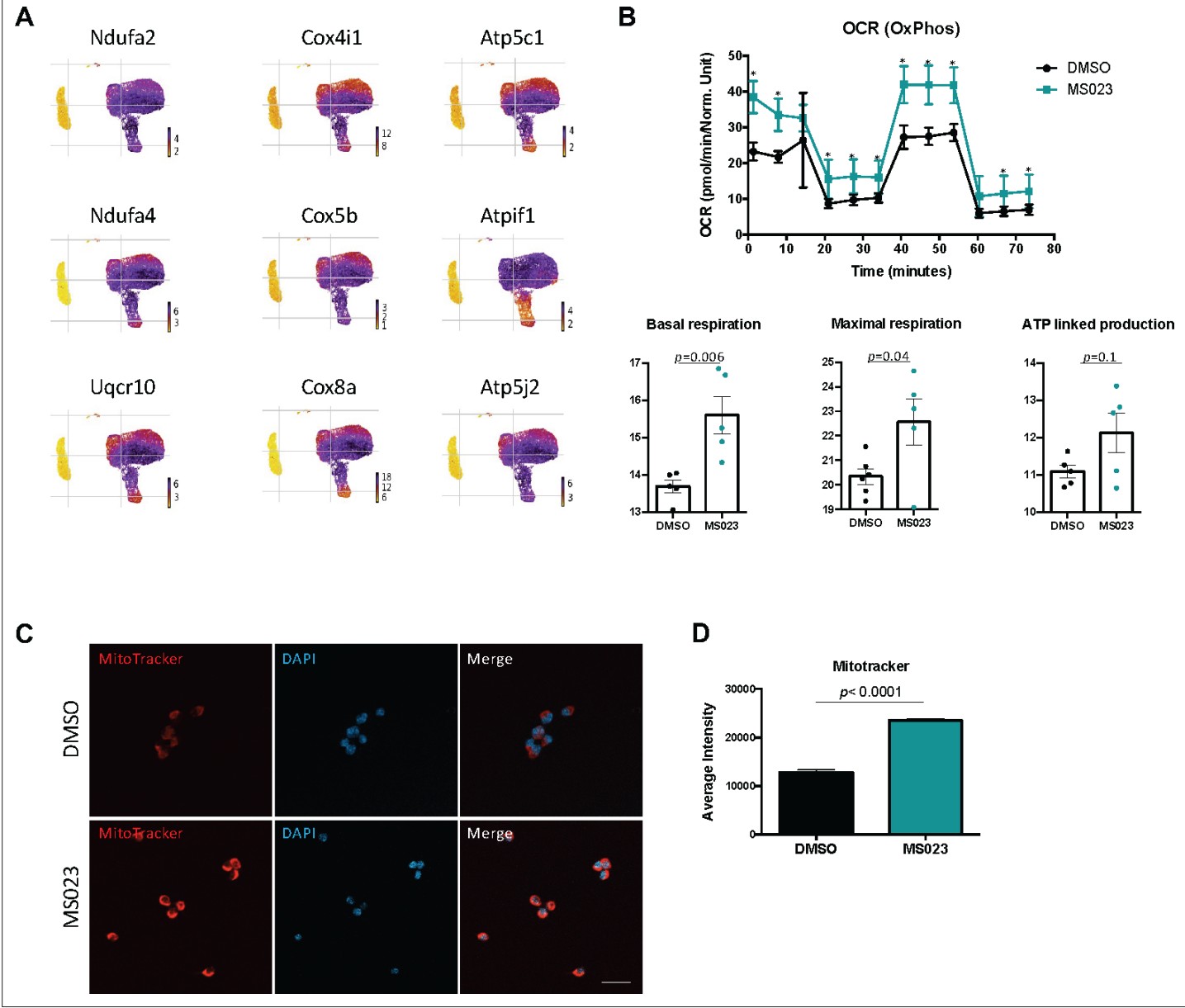

**Figure 4.** MS023-treated proliferating MuSCs display elevated oxidative phosphorylation. (**A**) UMAP plots showing expression of components of the electron transport chain localized to MS023 clusters. (**B**) Seahorse XF96 analysis of oxygen consumption rate (OCR) in DMSO and MS023-treated freshly isolated primary MuSCs and quantification of basal respiration, maximal respiration, and ATP-linked production. Error bars represent mean ± SEM from seven individual wells per condition. (**C**) MitoTracker staining (red) of freshly isolated MuSCs treated with DMSO (upper panel) or MS023 (lower panel) and counterstained with DAPI to visualize nuclei. Minimum 100 cells quantified per condition from three independent experiments. Scale bar represents 20 µM. (**D**) Quantification of MitoTracker signal intensity of cells from (**C**), error bars represent mean ± SEM from three individual replicates, >100 nuclei quantified per condition (p<0.0001).

The online version of this article includes the following figure supplement(s) for figure 4:

**Figure supplement 1.** Glycolysis is increased MS023-treated MuSCs.

## In vitro expanded MS023-treated MuSCs retain the potential to become mature myoblasts

We next wanted to validate that removal of MS023 created conditions which are permissive to terminal differentiation. Primary MuSCs were cultured under four conditions before switching to differentiation media: (1) for 4 days with DMSO, (2) washout for additional 2 days without DMSO (DMSOwo), (3) for 4 days with MS023, and (4) washout for additional 2 days without MS023 (MS023wo). Cells treated

for 4 days were switched to differentiation media with the continued presence of DMSO or MS023. Washout cells were switched to differentiation media without DMSO or MS023. Expectedly, cells differentiated in the presence of MS023 had an impaired ability to form multinucleated myotubes and had a poor fusion index of 19% compared to the 77% fusion index observed with DMSO treated MuSCs (*Figure 5A and B*). Interestingly, MS023wo cells had completely restored their ability to terminate differentiation, with their fusion index of 76% being comparable to that of DMSOwo cells (78%) (*Figure 5C and D*). Together, these experiments define MS023 as a reversible inhibitor of type I PRMT activity which enhances MuSC proliferation and self-renewal while still allowing for full differentiation capabilities upon removal of the compound.

## Type I PRMT inhibitor MS023 enhances MuSC engraftment

We next sought to confirm that MS023 washout conditions create a pool of MuSCs that are regeneration-competent upon engraftment into an injured muscle in vivo. Primary green fluorescent protein (GFP) expressing MuSCs were purified from a β-actin-GFP reporter mouse (C57BL/6-Tg(CAG-EGFP)10 sb/J) using an FACSAriaIII cell sorter (*Figure 5—figure supplement 1B and C*), and treated with MS023 or DMSO in vitro for 6 days to allow their expansion. Following culturing, 15,000 GFP⁺ myoblasts from each group were injected into the tibialis anterior (TA) muscle of wild type mice. One day after stem cell injection, the TA was injured with cardiotoxin (CTX). After 3 weeks, the mice were sacrificed and GFP⁺ differentiated muscle fibers were quantified following cross-sectional staining of the injured TA muscle (n=3 mice per condition) (*Figure 5—figure supplement 1A*). Interestingly, the mice that received MS023-treated MuSCs had more GFP⁺ mature muscle fibers than the mice that received DMSO-treated MuSCs (88 fibers/section ±32 for MS023 versus 31 fibers/section ±14 for DMSO, p=0.02; *Figure 5E and F*). These data indicate that MS023-treated MuSCs were able to differentiate in vivo following type I PRMT withdrawal and contribute efficiently to muscle regeneration following injury.

We next wanted to determine whether MS023-treated engrafted cells were able to repopulate the niche. Therefore, we repeated the engraftment experiment with MS023 and DMSO-treated GFP⁺ MuSCs, and this time re-isolated MuSCs 3 weeks after CTX injection. Strikingly, MS023-treated MuSC engraftment yielded an ~7-fold increase in the number of GFP⁺ following muscle regeneration (p=0.004; *Figure 5G and H*).

## MS023 injection increases muscle strength in *mdx* mice

Patients with DMD lack functional dystrophin, a structural protein required to connect mature myofibers to the extracellular matrix. The resulting muscle tissue is vulnerable to injury and is caught in constant cycles of muscle degeneration and regeneration (*Serrano et al., 2011*). The *Dmdᵐᵈˣ (mdx)* mice harbor a spontaneous mutation in the *dystrophin* gene and are therefore commonly used as a model for DMD (*Bulfield et al., 1984*; *Ryder-Cook et al., 1988*).

To determine whether MS023 could provide any therapeutic benefits in a dystrophic context, we delivered 80 mg/kg of MS023 or vehicle via intraperitoneal injection to *mdx* mice once a day for 3 days. Tail pieces were collected 48 hr after the final injection and whole tissue lysate was analyzed for ADMA- and SDMA-containing proteins using immunoblotting. We observed reduced ADMA proteins and a subsequent increase in SDMA proteins in MS023-treated *mdx* mice compared to vehicle-treated *mdx* mice (*Figure 6—figure supplement 1A*). The average ADMA relative expression was 0.95 for vehicle-treated mice and 0.83 for MS023 treated mice (p<0.00041). The average SDMA relative expression was 0.92 for vehicle-treated mice and 0.94 for MS023 treated mice (p<0.17). These whole-body measurements as measured by tail biopsies show MS023 promotes the decrease of proteins with ADMA and a slight increase in proteins with SDMA. It is known that inhibition of type I PRMTs or PRMT1 deletion diminishes ADMA and increases SDMA due to substrate scavenging (*Dhar et al., 2013*). At 10 days following the final injection, 2-paw and 4-paw grip strength measurements were taken. Additionally, endurance was tested by placing mice on a wire grid and inverting them so that the amount of time spent hanging could be recorded (see *Figure 6A* for schematic). We observed a 50% increase in 2-paw grip strength (p=0.0113) and a 45% increase (p=0.0089) in 4-paw grip strength in the MS023-treated mice versus those that received vehicle at the 10-day time point (*Figure 6B*). The hanging test to measure endurance showed a positive trend towards increased endurance at the 10-day time point in MS023 versus vehicle-treated mice, however this was not

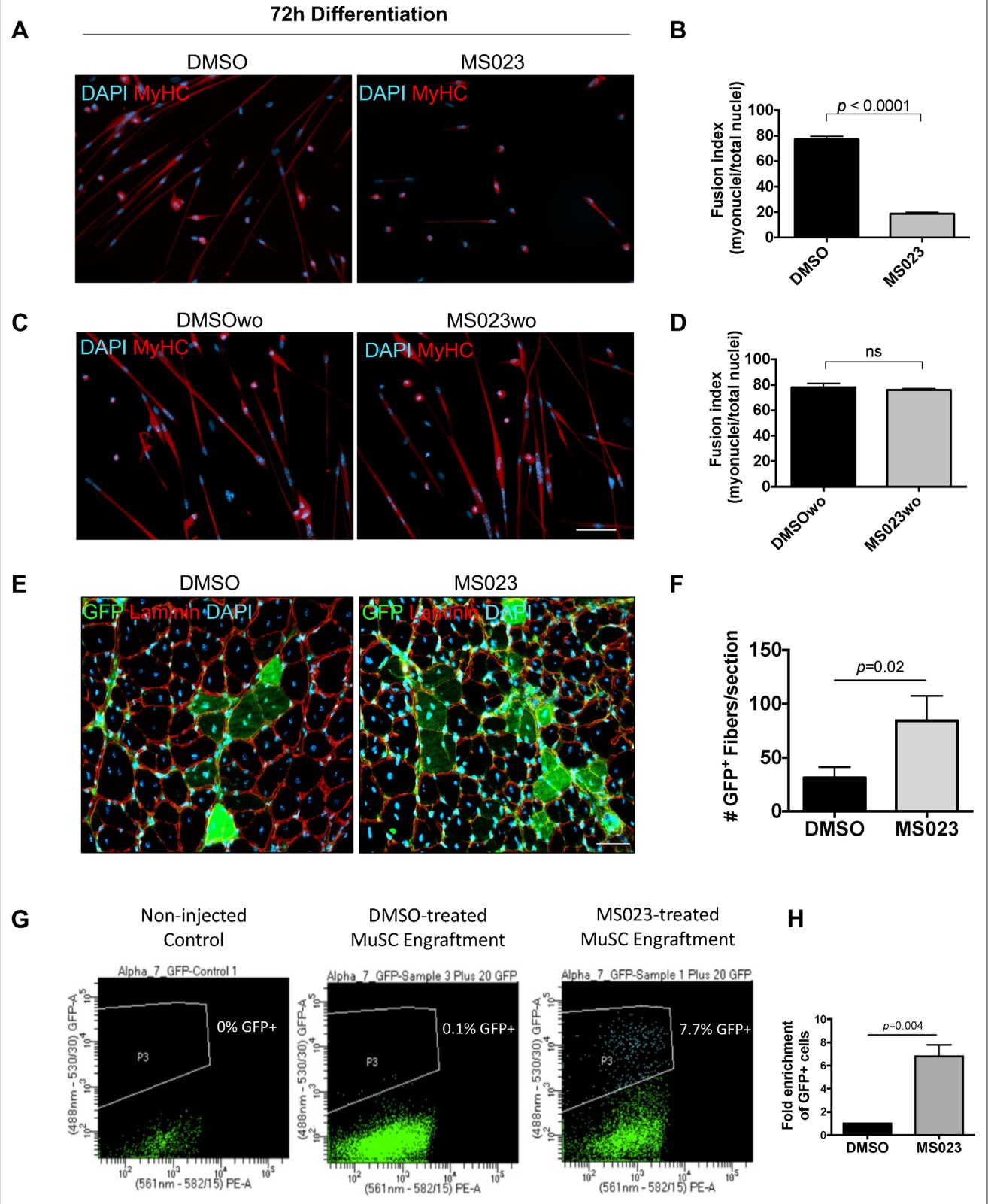

**Figure 5.** MS023 treatment is reversible and allows expanded MuSCs to differentiate ex vivo and in vivo. (**A**) Immunofluorescence of MyHC counterstained with DAPI in primary myotubes that were differentiated in the presence of DMSO or MS023 following 4 days of treatment with DMSO or MS023. (**B**) Fusion index calculated as the ratio of nuclei within multinucleated myotubes to total nuclei from >200 nuclei per condition, error bars represent mean ± SEM from three individual replicates. (**C**) Immunofluorescence of MyHC counterstained with DAPI in primary myotubes that were

*Figure 5 continued on next page*

*Figure 5 continued*

differentiated in the absence of DMSO or MS023 following 4 days of treatment with DMSO or MS023 and 2 days of washout. Scale bar represents 50 μM (**D**) Fusion index calculated the same as 6B. (**E**) Immunofluorescence of GFP⁺ myofibers following transplantation of 15,000 DMSO or MS023-treated MuSCs. Scale bar represents 50 μM. (**F**) Quantification of GFP⁺ myofibers. Error bars represent mean ± SEM from three individual biological replicates. (**G**) Representative FACS plots of re-isolated MuSCs following engraftment and muscle regeneration. P3 represents the population of purified GFP⁺ MuSCs. (**H**) Fold enrichment of MS023-treated MuSC re-isolation compared to DMSO. Error bars represent mean ± SEM from three individual biological replicates.

The online version of this article includes the following figure supplement(s) for figure 5:

**Figure supplement 1.** Type I PRMT inhibitor MS023 enhances MuSC engraftment.

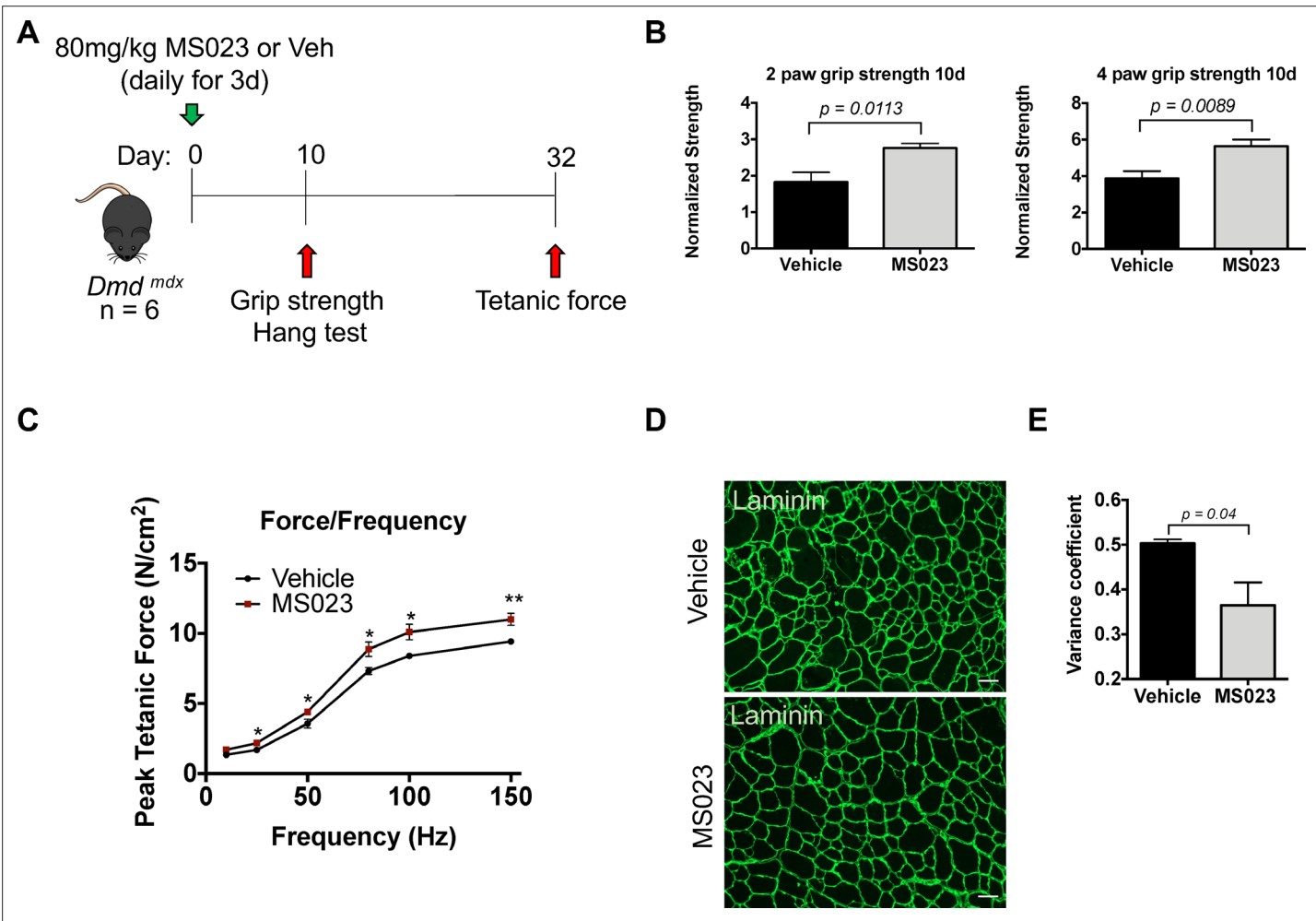

**Figure 6.** In vivo administration of MS023 to dystrophic mice improves muscle strength. (**A**) Experimental schema. (**B**) Normalized grip strength measurements taken from two forelimbs and all four limbs of mice treated with vehicle or MS023 10 days after the last injection. Error bars represent mean ± SEM from six biological replicates. (**C**) Force/frequency curve generated for mice treated with vehicle or MS023. (**D**) Representative cross-sectional area of TA muscles isolated from mice treated with vehicle or MS023 and immunostained with anti-laminin antibodies to visualize myofibers. Scale bar represents 50 μM. (**E**) Quantification of the variance coefficient of TA muscle minimum fiber Feret measurement from mice treated with vehicle or MS023. Error bars represent mean ± SEM from three biological replicates, >200 fibers measured per mouse.

The online version of this article includes the following source data and figure supplement(s) for figure 6:

**Figure supplement 1.** MS023 decreases ADMA containing proteins globally and increases strength in *mdx* mice.

**Figure supplement 1—source data 1.** Uncropped immunoblot.

statistically significant (p=0.1; *Figure 6—figure supplement 1B*). A final measurement was performed at the 32-day endpoint, during which mice were placed under terminal anesthesia and the extensor digitorum longus (EDL) hindlimb muscle was isolated, incubated ex vivo, and used to generate a force/frequency curve. Interestingly, this experiment revealed that MS023-treated mice responded with a~30% increase in force compared to their vehicle-treated counterparts (p=0.00727; *Figure 6C*). Moreover, we observed a reduction in the variation of the minimum Feret diameter measurements of fibers from TA cross-sections of MS023-treated mice (p=0.04; *Figure 6D and E*). These findings show that injection of type I PRMT inhibitor MS023 in *mdx* mice improved muscle strength.

## Discussion

In the present manuscript, we show that inhibition of type I PRMTs promoted the in vitro proliferation of Pax7[+] MuSCs, as visualized by ki67 staining. scRNAseq analysis of MS023 treated MuSCs identified new transitional clusters. MS023-specific subpopulations harboured unique transcriptional signatures of stemness and energy metabolism. The MS023-treated MuSCs exhibited elevation of both glycolysis and OxPhos. Engrafted MuSCs treated with MS023 were efficient at regenerating injured muscle and at repopulating the niche. Treatment of the *mdx* mouse model of DMD with MS023 resulted in enhanced grip strength and force generation. Repression of type I PRMTs altered the metabolic state of cultured MuSCs while maintaining their capacity for self-renewal and to effectively participate in muscle regeneration. These findings suggest that type I PRMT inhibitors may have therapeutic potential for treating certain skeletal muscle diseases.

The ability to tailor energy metabolism to the specific demands of cell proliferation and differentiation is known as metabolic plasticity, and is required for maintaining tissue homeostasis (*Folmes et al., 2012*). The transition from quiescence to proliferation is regulated by a shift from OxPhos to glycolysis (*Pala et al., 2018*; *Ryall et al., 2015*). During MuSC quiescence, high levels of NAD +are generated by oxidative metabolism which in turn activates Sirtuin 1 (SIRT1) to deacetylate and repress expression of differentiation genes required for the myogenic program (*Ryall et al., 2015*). The shift from OxPhos to glycolysis upon exit from quiescence results in reduced NAD +levels and subsequent activation of differentiation genes. Furthermore, it has been shown that entry to the process of terminal differentiation is mediated by the activation of pyruvate dehydrogenase (PDH), marking a switch from glycolysis to OxPhos (*Hori et al., 2019*). Another key player in metabolic plasticity in MuSCs is AMP kinase, a major energy sensor in the cell, regulating energy balance and metabolic state (*Hardie et al., 2016*; *Herzig and Shaw, 2018*). Null alleles of AMPK have contributed to the understanding of its role in MuSCs (*Fu et al., 2015b*; *Theret et al., 2017*). Activated AMPK stimulates autophagy, mitophagy, and mitochondrial biogenesis, thus maintaining mitochondrial homeostasis (*Herzig and Shaw, 2018*). Our data showed that inhibition of type I PRMTs in MuSCs stimulated the expression of autophagic markers (*Atg5, Lamp1, Lamp2, Sqstm1*) downstream targets of AMPK, however it remains to be determined whether AMPK is involved in MS023-treated cells.

PRMT1 has been shown to interact with AMPK in skeletal muscle during the early stages of muscle atrophy (*Stouth et al., 2018*). Moreover, CARM1 directly methylates AMPK to regulate its activity during muscle response to denervation (*Stouth et al., 2020*). PRMT1 has also been shown to play a signaling role to regulate thermogenesis. PRMT1 acts through the transcriptional co-activator PGC-1αin human and mouse adipocytes to regulate thermogenic fat activation, and PRMT1-deficient mice were unable to induce the thermogenic program following cold exposure (*Qiao et al., 2019*). Methylation of PGC-1αby PRMT1 was also shown to stimulate mitochondrial biogenesis and may regulate nuclear-encoded mitochondrial genes (*Teyssier et al., 2005*). Additionally, PRMT1-mediated methylation positively regulates the insulin receptor (IR)-phosphatidylinositol 3-kinase (PI3-K) pathway which is required for glucose transport in skeletal muscle (*Iwasaki and Yada, 2007*). Overexpression of PRMT1 in the liver under hypoxic conditions causes hypermethylation of FoxO1 and increased translocation to the nucleus, resulting in reduced glucose uptake (*Bayen et al., 2018*). Therefore, these findings indicate that PRMT1 methylates key regulators of cellular metabolism and highlight a crosstalk between metabolic effectors and type I PRMTs.

Treatment of MuSCs with MS023 resulted in metabolic reprogramming of MuSCs, supporting a role for type I PRMTs as metabolic regulators. In vitro, MS023 has been shown to inhibit several type I PRMTs at nanomolar (nM) concentrations (*Eram et al., 2016*). It is well-documented that PRMT1 is the major cellular type I enzyme (*Pawlak et al., 2000*) and this is why PRMT1, but not the other

type I PRMTs are embryonic lethal in mice (*Guccione and Richard, 2019*). The numerous published data *in cellulo* with MS023 are thus far only reproduced by PRMT1-deficiency by siRNA or knockout, suggesting that MS023 actions in vivo are predominantly mediated by inhibiting PRMT1 (*Gao et al., 2019*; *Plotnikov et al., 2020*; *Wu et al., 2022*; *Zhu et al., 2019*). In addition, PRMT1 is by far the most expressed type I PRMT in MuSCs at every time point tested in skeletal muscle post-cardiotoxin injury (uninjured, and 0.5, 2, 3.5, 5, 10, 21 days post-injury) by single cell RNAseq (*Oprescu et al., 2020*). Thus, the effects of MS023 on MuSCs are most likely mediated by inhibition of PRMT1.

scRNAseq analysis of MS023-treated MuSCs identified new transitional clusters which harboured transcriptional signatures of stemness and energy metabolism. DMSO clusters M1/M2/M4 and MS023 clusters M3/M5 were both enriched for proliferation markers (*Cdk1, Mki67*), and M3/M5 also uniquely harboured elevated metabolic genes including components of the electron transport chain (*Atp5k, Cox5a, Ndufa2*) and components of glycolysis (*Eno1, Gapdh*). The simultaneous elevation of both glycolytic and OxPhos components in MS023 subpopulations likely contributes to the enhanced proliferation phenotype. MuSC subpopulations have been characterized at the single-cell level during muscle regeneration, revealing regulatory mechanisms that guide the transition from quiescence to activation, proliferation, and differentiation. A general progressive elevation of metabolic enzyme expression has been noted from low levels in MuSCs isolated from uninjured muscle, to slightly elevated levels in MuSCs isolated from acutely injured muscle (60 hr after injury), to the highest levels in cultured proliferating MuSCs (*Dell'Orso et al., 2019*). A more in-depth analysis of MuSC subpopulations throughout muscle regeneration investigated additional time points after injury (2, 5, and 7 days) and generated a hierarchical continuum model of MuSCs throughout the regeneration process, with a focus on ligand-receptor cell communication networks (*De Micheli et al., 2020*). Additionally, analysis of young and aged MuSCs revealed that, while they have similar transcriptional signatures, aged MuSCs display delayed activation state dynamics when exiting quiescence, thus preventing them from entering the differentiation program as efficiently as young MuSCs (*Kimmel et al., 2020*). It has been previously reported that the quiescent MuSCs captured by scRNAseq exist in two groups, one 'quiescent' and one 'early activation', wherein the early activation group highly expresses *Fos* and *Jun* while down-regulating *Hox* genes (*van den Brink et al., 2017*). Our Q1 and Q2 clusters are in alignment with these two states (Q1 - quiescence; Q2 - early activation), and bare transcriptional similarities to our MS023-specific clusters M3 and M5, such as upregulated *Fos* and *Jun*, as well as high levels of *Pax7*. Notably, capture of rare transient MuSC states remains difficult due to poor sampling or limited cell number. To overcome these issues, one study evaluated 365,000 single cells/nuclei from over 100 combined datasets, allowing for the characterization of short-lived MuSC subpopulations that emerge throughout the myogenic process (*McKellar et al., 2021*). We show that MS023 treatment generates subpopulations of self-renewing MuSCs that have not been previously identified, suggesting that transient cell states may be also amplified through treatment with epigenetic inhibitors, thus facilitating a more in-depth understanding of their function and contributions to muscle regeneration.

PRMTs are known as epigenetic regulators by methylating histones directly and by serving as coactivators/repressors (*Xu and Richard, 2021*). We now show MuSC cells treated with type I PRMT inhibitors have altered metabolism favouring proliferation and self-renewal, however, the mechanism whether epigenetic, regulating of mRNA splicing or a signaling cascade remains to be defined. It has been shown that inhibition of the epigenetic modulator *Setd7* enhances the proliferation of cultured MuSCs, while retaining their stemness by preventing transport of β-catenin into the nucleus, thereby failing to activate the differentiation program (*Judson et al., 2018*). Taken together, these findings suggest that inhibiting methyltransferases can affect MuSC fate and perhaps both lysine (Setd7) and arginine (MS023) inhibitors might provide a more favorable combination to promote the expansion of MuSCs, while maintaining their stem-cell-like properties.

We report heightened engraftment capabilities of MS023-treated MuSCs accompanied by high Pax7 expression levels. It has been shown that freshly isolated MuSCs retain high Pax7 expression levels and are superior at engrafting compared to cultured MuSCs (*Montarras et al., 2005*; *Sacco et al., 2008*). Generating stable Pax7 expression in myogenic precursor cells with lentiviral delivery of transgenes is a strategy for enhancing engraftment potential following ex vivo expansion of MuSCs (*Kim et al., 2021*), although this approach may be hampered by safety concerns. Therefore, we have identified an additional strategy for increasing Pax7 expression and improving

engraftment efficiency and self-renewal of ex vivo expanded MuSCs through inhibition of type I PRMTs. Furthermore, our findings show that injection of MS023 in the dystrophic mouse *mdx* model led to enhanced muscle strength with effects lasting up to 30 days. We cannot exclude if the effect of MS023 was mediated by an expansion of the MuSC pool or by an effect on other cell types, such as a direct impact on the myofibers. The goal of this experiment was to provide a therapeutic perspective for the possible use of type I PRMT inhibitor for the treatment of DMD. Our findings suggest that type I PRMT inhibitors may have therapeutic potential for treating certain skeletal muscle diseases. For instance, to improve the efficacy of autologous cell therapy, the dystrophin-deficient MuSCs collected from DMD patient and corrected by CRISPR prime editing (*Happi Mbakam et al., 2022*) could be treated with MS023 to maintain their stemness and enhance their cell engraftment capacity.

In sum, our findings show that inhibition of type I PRMTs increased the proliferation capabilities of MuSCs with altered cellular metabolism, while maintaining their stem-like properties such as self-renewal and engraftment potential.

## Materials and methods
### Mice
C57BL/6 J (Jackson Laboratory 000664) were the wild type mice used for MuSC FACS purification for scRNAseq, muscle fiber isolation, and for all other in vitro experiments.

β-Actin GFP mice (C57BL/6-Tg(CAG-EGFP)10 sb/J, Jackson Laboratory 003291) were used to obtain GFP⁺ MuSCs for further implantation experiments. Mice lacking dystrophin expression (C57BL/10ScSn-*Dmd^mdx*/J, Jackson Laboratory 001801) were used for intraperitoneal MS023 injection and muscle physiology assessment. The experiments with the *Dmd^mdx* mice were performed at Sainte-Justine and were approved by the CHU Sainte-Justine Research Ethics Committee and performed in compliance with the Comité Institutionnel des Bonnes Pratiques Animales en Recherche (CIBPAR; approval number 2020–2668) in accordance with the Canadian Council on Animal Care guidelines. All other mouse husbandry and experiments were conducted in accordance with the Institutional Animal Care and Use Committee (IACUC) of McGill University. All animal procedures conducted at McGill were approved by the Animal Welfare Committee of McGill University (protocol #3506).

### Primary MuSC isolation
Skeletal muscle tissue was isolated from the abdominal and diaphragm muscles of wild type mice and muscle stem cells were isolated as previously described using fluorescence activated cell sorting (FACS) (*Pasut et al., 2012*). Briefly, dissected muscles were minced with dissection scissors and digested with collagenase/dispase solution (2.4 U/ml collagenase D, 2.4 U/ml Dispase II in Ham's F10 media) at 37 °C for 1 hr. Digested tissue was triturated and filtered through a 40 µM cell strainer. Cells were pelleted for 18 min at 1800 rpm and resuspended in 2% BSA/PBS. Cells were stained for 15 min at room temperature with ITGA7-Alexa647 (R&D systems) for positive selection, and PE-CD45 (Invitrogen), PE-CD11b (Invitrogen, PE-CD31 (BD Pharmigen), and PE-ScaI (BD Pharmigen)) for negative selection. Hoescht was used to gate the living cells. Cells were washed once with 2% BSA/PBS and pelleted prior to final resuspension and one last filter through a 40 µM cell strainer. ITGA7+/CD45-/CDCD11b-/Sca1-/Hoechst + cells were sorted into full myoblast growth media using the FACSAriaIII cell sorter (BD Biosciences).

### Myoblast growth and differentiation culture
Purified MuSCs were seeded onto collagen-coated plates and expanded in growth media (Ham's F10 media with 20% FBS, 2.5 ng/mL human recombinant bFGF, and 1% Penicillin/Streptomycin) at 37 °C and 5% $CO_2$. Media was replenished every two days. To differentiate myoblasts into myotubes, myoblasts were grown to 90% confluency in growth media, washed twice with 1 X PBS, and switched to differentiation media (DMEM, 1% FBS, and 1% Penicillin/Streptomycin). For inhibitor treatment, cells were incubated with 0.033% DMSO or 1 µM MS023 (Sigma) in 0.033% DMSO and media was replenished every second day for the duration of the indicated treatment time.

## Western blot

Proteins from total cell or tissue lysate (50 mM HEPES [pH 7.4], 150 mM NaCl, and 1% Triton X-100) were resolved on SDS 8–15% polyacrylamide gels and transferred onto nitrocellulose membranes using the Trans-Blot turbo transfer system (BioRad). Membranes were blotted with the primary antibodies against ADMA (Epicypher, SKU: 13–0011), SDMA (Millipore Cat# 07–413, RRID:AB_310595), and β-Actin (Sigma-Aldrich Cat# A2228, RRID:AB_476697) overnight at 4 °C. Following three washes in TBST, membranes were incubated with HRP-conjugated secondary antibodies (Sigma) for 45 min and visualised on X-ray films with Western Lightning Plus ECL (Perkin Elmer).

## Cultured myoblast immunofluorescence

Myoblasts expanded in growth media were seeded onto collagen-coated glass coverslips (VWR) in 6-well plates. Coverslips were transferred to a 12-well plate containing 4% paraformaldehyde (PFA) and cells were fixed for 15 min at room temperature. Cells were then permeabilized with 0.2% Triton X-100, 0.125 M glycine in PBS for 12 min at room temperature (RT). Incubation with blocking buffer (2% BSA, 5% horse serum and 0.1% Triton X-100) proceeded for 1 hr at RT. Primary antibodies were diluted in blocking buffer to detect Pax7 (Developmental Studies Hybridoma Bank, 1:100), MyoD (Santa Cruz Biotechnology, 1:200), or ki67 (Abcam, 1:1000). After 16 hr incubation at 4 °C, cells were washed 3 times with 1 X PBS for 10 min. Secondary antibody (AlexaFluor anti-mouse or anti-rabbit 488 nm or 568 nm) was used at a dilution of 1:400 in blocking buffer for 45 min in the dark at RT. Cells were washed 3 times for 10 min with 1 X PBS. Finally, the cover slips were transferred to a microscope slide and mounted with ProLong Gold Antifade Mountant with DAPI (ThermoFisher Scientific). For MitoTracker staining, primary MuSCs were seeded onto 8-well Lumox chamber slides (Sarstedt) and cultured for the indicated time points. Cells were then incubated for 30 min with a final concentration of 100 nM MitoTracker Red CMXRos (Thermo Fisher Scientific) at 37 °C and 5% $CO_2$. Cells were fixed with ice-cold 100% methanol at –20 °C for 15 min and washed 3 times for 10 min with 1 X PBS prior to mounting with ProLong Gold Antifade Mountant with DAPI. All cells were visualized on a Zeiss Axio Imager M1 microscope (Carl Zeiss, Thornwood NY), and resulting images were analyzed using Zeiss' ZEN Digital imaging suite software.

## Muscle fiber isolation and culture

Wild type mice were sacrificed and their extensor digitorum longus (EDL) muscles were dissected using standard dissection techniques. Isolated muscles were incubated with 0.4% collagenase (Sigma) in DMEM for 30 min at 37 °C and 5% $CO_2$. Whole muscle was then triturated with a plastic disposable Bohr pipette to dissociate individual fibers from the whole muscle as described previously (*Gallot et al., 2016*). To mimic activating conditions, fibers were cultured in fiber growth media (DMEM plus 20% FBS, 1% chick embryo extract, 2.5 ng/mL bFGF, 1% penicillin/streptomycin) at 37 °C and 5% $CO_2$. For quiescent satellite cell analysis, fibers were fixed immediately following dissociation using 4% PFA prepared fresh in 1 X PBS.

## Muscle fiber immunofluorescence

Cultured muscle fibers from wild type mice were fixed in 4% paraformaldehyde PFA and washed twice with 1 X PBS. Fibers were then permeabilized with 0.2% Triton X-100, 0.125 M glycine in PBS for 15 min at RT. Blocking followed for 1 hr at room temperature with 2% BSA, 5% horse serum and 0.1% Triton X-100. Primary antibodies were diluted in blocking buffer to detect Pax7 (Developmental Studies Hybridoma Bank, 1:100), MyoD (Santa Cruz Biotechnology, 1:200), ki67 (Abcam, 1:1000). After 16 hr incubation at 4 °C, fibers were washed 3 times with 1 X PBS for 10 min. Secondary antibody (AlexaFluor anti-mouse or anti-rabbit 488 nm or 568 nm) was used at a dilution of of 1:400 in blocking buffer for 45 min in the dark at RT. Fibers were washed three times for 10 min with 1 X PBS. Finally, the fibers were transferred to a microscope slide outlined using an ImmEdge hydrophic barrier pen and mounted with ProLong Gold Antifade Mountant with DAPI (Thermo Fisher Scientific). Fiber-associated satellite cells were then visualized on a Zeiss Axio Imager M1 microscope (Carl Zeiss, Thornwood NY), and resulting images were analyzed using Zeiss' ZEN Digital imaging suite software.

## scRNAseq sample preparation and computational analysis

Each biological replicate for the scRNAseq corresponded to one C57BL/6 mouse. The purified MuSCs were cultured in Ham's F10 (Gibco) with 20% FBS (HyClone), 2.5 ng/mL human recombinant bFGF (Gibco), 1% Penicillin/Streptomycin (Wisent Inc) with 0.033% DMSO or 1 μM MS023 in 0.033% DMSO. The medium was changed on day 2 and day 4 of culturing. The purified MuSCs from each replicate were stained for viability with calcein-AM and ethidium-homodimer1 (P/ N L3224 Thermo Fisher Scientific). scRNA libraries were generated using the GemCode Single- Cell Instrument (10 x Genomics, Pleasanton, CA, USA) and Single Cell 3' Library & Gel Bead Kit v2 and ChIP Kit (P/N 120236 P/N 120237 10 x Genomics). The sequencing ready libraries were purified with SPRIselect, quality controlled for sized distribution and yield (LabChip GX Perkin Elmer), and quantified using qPCR (KAPA Biosystems Library Quantification Kit for Illumina platforms P/N KK4824) as previously described (*Couturier et al., 2020*). Libraries were subsequently shipped and sequenced using Illumina NovaSeq6000 at IGM Genomics Center, UCSD, San Diego, CA. Cell barcodes and UMI (unique molecular identifiers) barcodes were demultiplexed and paired-end reads were first aligned to the mouse genome (mm10) using the Cell Ranger software v3.1.0 (10X Genomics, https://support.10xgenomics.com/single-cell-gene-expression/software/pipelines/latest/what-is-cell-ranger). Pre-processing was then carried out with the Seurat v3.2.0 R package (*Butler et al., 2018*). Genes detected in less than 3 cells as well as cells containing less than 200 genes detected were removed. Cells were further filtered out for each sample based on the distribution of genes detected as well as the percentage of mitochondrial counts to balance the number of cells per sample to ~4000, and the raw count matrices of all samples were merged. Read counts for each cell were then normalized by the cell total, multiplied by 10,000 and natural-log transformed. The expression of the 2000 genes with highest cell-to-cell variation was standardized and the heterogeneity associated with mitochondrial contamination was regressed out. Principal component analysis was performed on the scaled data and the top 10 principal components were used to construct a K-nearest neighbor cell graph. Clustering of cells was carried out through the Louvain algorithm with the granularity parameter set to 0.4 and visualized with the Uniform Manifold Approximation and Projection (UMAP) (*Becht et al., 2018*) dimensional reduction technique using the Scanpy v1.5.2 python module (). Cluster biomarkers were identified using the Wilcoxon Rank Sum test. Cell trajectories across pseudotime were analyzed by the Monocle v2.16.0 R package (*Trapnell et al., 2014*). Cell progress was defined by differentially expressed genes based on the clusters identified by Seurat. The dimensionality of the data was reduced through the Discriminative Dimensionality Reduction with Trees (DDRTree) algorithm to two dimensions and the cells were ordered along the trajectory according to pseudotime. Genes with branch-dependent expression were identified through the branched expression analysis modeling (BEAM) test. RNA velocity analysis was performed by the scVelo v0.2.2 python module (). Spliced and unspliced mRNAs were first distinguished through the velocyto v0.17.17 python module (). Velocities representing the direction and speed of cell motion were then computed and projected onto the UMAP embedding.

## Oxygen consumption rate and extracellular acidification measurements

Oxygen consumption rates (OCR) and extracellular acidification rate (ECAR) were measured using a Seahorse XFe96 Flux Analyzer and analyzed with Wave 2.6.0 software (Agilent Technologies). FACS sorted muscle stem cells were seeded in a matrigel-coated 96-well plate and cultured with MS023 or DMSO for 48 hr. Cells were incubated in MitoStress test assay media (DMEM 5030 media, 10 mM glucose, 1 mM sodium pyruvate, 2 mM glutamine, PH7.4) or GlycoStress test assay media (DMEM 5030 media, 2 mM glutamine, pH 7.4) 1 hr prior to OCR and ECAR measurements, respectively. Oligomycin (3 mM, Sigma-Aldrich), carbonyl cyanide-p-trifluoromethoxy-phenylhydrazone (FCCP; 1 mM, Abcam) and rotenone-antimycin A (RAA; 2.5 mM, Sigma-Aldrich) were added to cells to measure OCR parameters while glucose (10 mM, Sigma-Aldrich), Oligomycin (1 mM, Sigma-Aldrich) and 2-DG (100 mM) were used to measure ECAR parameters. Following the assay, cells were stained with crystal violet and nuclei were counted in each well for normalization.

## MuSC engraftment and muscle histology

Primary MuSCs were FACS-purified from donor GFP mice and cultured with MS023 or DMSO for 5 days. Cells were then trypsinized, resuspended in 1 X PBS, and pelleted for 10 min at 1800 RPM. Cell pellets were resuspended in 1 ml of 1 X PBS and counted using a haemocytometer. A total of

15,000 cells from each condition were then injected into the TA muscle of wild type mice. Recipient mice were also injected with 50 µl of 10 µM cardiotoxin (CTX) 24 h prior to MuSC engraftment to induce muscle regeneration. Recipient mice were sacrificed 3 weeks after CTX injection, and the TA muscles were dissected and fixed in 2% PFA at 4 °C for 16 hr. Fixed TA muscles were then embedded in OCT and frozen prior to cryosectioning onto a glass microscope slide. Resulting tissue sections were permeabilized for 12 min with 0.2% Triton X-100, 0.125 M glycine in PBS at RT. Blocking followed with M.O.M blocking reagent (Vector Laboratories) for 1 hr at RT. Primary antibodies were diluted in blocking reagent to detect Pax7 (Developmental Studies Hybridoma Bank, 1:10) and laminin (Sigma, 1:50). After 16 hr incubation at 4 °C, sections were washed 3 times with 1 X PBS for 10 min. Secondary antibody (AlexaFluor anti-mouse or anti-rabbit 488 nm or 568 nm) was used at a dilution of 1:400 in blocking reagent for 45 min in the dark at RT. Sections were then washed three times for 10 min with 1 X PBS. Tissue sections were mounted with ProLong Gold Antifade Mountant with DAPI (ThermoFisher Scientific) and covered with a coverslip. Visualization was performed on a Zeiss Axio Imager M1 microscope (Carl Zeiss, Thornwood NY), and resulting images were analyzed using Zeiss' ZEN Digital imaging suite software.

## MS023 injection in mdx mice

12-week-old *mdx* mice received an intraperitoneal injection of 80 mg/kg MS023 (Sigma) dissolved in 50 µl N-methyl-2-pyrrolidone, 200 µl Captisol, 200 µl polyethylene glycol 400 and 550 µl PBS once per day for three consecutive days. As control, 12-week-old mice received vehicle (50 µl N-methyl-2-pyrrolidone, 200 µl Captisol, 200 µl polyethylene glycol 400 in 550 µl PBS) alone. Tail pieces were collected 48 h after the last injection and lysed for western blot analysis with anti-ADMA and anti-SDMA and β-actin antibodies. Grip strength and hanging test measurements were taken at 10 days after the final injection. Twenty-nine days after the final injection, mice were anesthetized and sacrificed for endpoint ex vivo force measurements.

## Grip strength measurements and hanging test

Grip strength was measured for the front two paws and for all four limbs using a Bioseb Grip Strength Test instrument. Each mouse was placed on the rod (2 paws) or mesh (4 paws) and pulled from the tail three times with 1 min rest between trials. The highest measurement was retained for subsequent analysis. To assess endurance, mice were placed on a plastic mesh taped to a hollow cylinder. The cylinder was slowly inverted over soft bedding to allow mice to grip the mesh, and the amount of time each mouse remined suspended before falling onto the bedding was recorded 3 times with 2 min rest between trials. The longest time measurement was retained for analysis.

## Ex vivo force measurements of mdx mice

Mice were anesthetized with pentobarbital sodium (50 mg/kg). Proximal and distal tendons of the EDL were attached with silk suture (3.0) and the muscle was carefully isolated and placed in a buffered physiological solution (Krebs-Ringer supplemented with glucose and bubbling carbogen gas) maintained at 25 °C (*Dort et al., 2021*). The distal tendon of the EDL muscle was attached to the electrode, and the proximal tendon was attached to the force lever arm (300C-LR dual-mode lever; Aurora Scientific, Canada). Muscle length was adjusted to find the optimal muscle length ($L_0$) at which it generates the maximum isometric twitch tension. The EDL muscle was allowed to equilibrate in the contractile bath for 10 min equilibration prior the contractile measurement. Muscle was stimulated at different frequencies (10, 25, 50, 80, 100, 150 Hz), with 2 min rest between each stimulation, to obtain a force-frequency curve. Thereafter, muscle length and weight were measured. Specific muscle force ($N/cm^2$) was determined using the following formula: (force (N) x fiber length (0.44 x $L_0$ for the EDL muscle) x muscle density (1.06 $g/cm^3$))/muscle mass (g).

## Acknowledgements

The authors thank Chris Young and Mathew Duguay at the LDI Flow Cytometry Facility for their assistance with FACS experiments. We thank Drs. Vahab Soleimani, Vladimer Ljubicic and Etienne Audet-Walsh for critically reading the manuscript and helpful suggestions and discussions. The research was funded by Canadian Institute of Health Research (CIHR) FDN-154303 to SR. CD held a studentship from the fonds de la recherche en santé du Québec (FRQS).

## Additional information

### Funding

| Funder | Grant reference number | Author |
|---|---|---|
| Canadian Institutes of Health Research | FDN-154303 | Claudia Dominici |

The funders had no role in study design, data collection and interpretation, or the decision to submit the work for publication.

### Author contributions

Claudia Dominici, Conceptualization, Data curation, Formal analysis, Methodology, Writing – original draft, Project administration, Writing – review and editing; Oscar D Villarreal, Data curation, Project administration, Formal analysis, Writing - review and editing, Methodology, Writing – original draft; Junio Dort, Emilie Heckel, Yu Chang Wang, Data curation, Formal analysis, Methodology; Ioannis Ragoussis, Jean-Sebastien Joyal, Supervision, Methodology, Writing – original draft; Nicolas Dumont, Conceptualization, Formal analysis, Supervision, Methodology, Writing – original draft, Project administration, Writing – review and editing; Stéphane Richard, Conceptualization, Resources, Supervision, Funding acquisition, Methodology, Writing – original draft, Project administration, Writing – review and editing

### Author ORCIDs

Stéphane Richard ⓘ http://orcid.org/0000-0003-2665-4806

### Ethics

The experiments with the Dmdmdx mice were performed at Sainte-Justine and were approved by the CHU Sainte-Justine Research Ethics Committee and performed in compliance with the Comité Institutionnel des Bonnes Pratiques Animales en Recherche (CIBPAR; approval number 2020-2668) in accordance with the Canadian Council on Animal Care guidelines. All other mouse husbandry and experiments were conducted in accordance with the Institutional Animal Care and Use Committee (IACUC) of McGill University. All animal procedures conducted at McGill were approved by the Animal Welfare Committee of McGill University (protocol #3506).

Reviewer #1 (Public Review): https://doi.org/10.7554/eLife.84570.3.sa1
Reviewer #2 (Public Review): https://doi.org/10.7554/eLife.84570.3.sa2
Reviewer #3 (Public Review): https://doi.org/10.7554/eLife.84570.3.sa3
Author Response: https://doi.org/10.7554/eLife.84570.3.sa4

## Additional files

### Supplementary files

• Supplementary file 1. Expressed genes per cluster. Quiescence 1 and 2 (Q1 and Q2), myoblast clusters 1–5 (M1-M5), and differentiated myoblasts 1 and 2 (DM1 and DM2) clusters.

• Supplementary file 2. Top 100 genes enriched per cluster of MuSCs purified from 8-week-old C57BL/6 mice immediately after (1) isolation (termed d0; sample day 0), and (2) culture in growth medium for 4 days with 0.033% DMSO as control (sample d4) or with 1 µM MS023 (sample d4MS023), and (3) grown in growth media for 6 days with 0.033% DMSO removed at day 4 (sample d6), or 6 days in culture with 1 µM MS023 removed at day 4 (sample d6MS023rd4).

• Supplementary file 3. Sample distribution across clusters. The number of cells and the percentage distributed within clusters Q1, Q2, C1, M1, M2, M3, M4, M5, DM1 and DM2 is indicated.

• MDAR checklist

### Data availability

The single cell data is available in the GEO database under accession code GSE199420.

The following datasets were generated:

| Author(s) | Year | Dataset title | Dataset URL | Database and Identifier |
| --- | --- | --- | --- | --- |
| Richard S | 2023 | Day 0 | https://ncbi.nlm.nih.gov/geo/query/acc.cgi?acc=GSM5972490 | NCBI Gene Expression Omnibus, GSM5972490 |
| Richard S | 2023 | Day 4 DMSO | https://www.ncbi.nlm.nih.gov/geo/query/acc.cgi?acc=GSM5972491 | NCBI Gene Expression Omnibus, GSM5972491 |
| Richard S | 2023 | Day 4 MS023 | https://www.ncbi.nlm.nih.gov/geo/query/acc.cgi?acc=GSM5972492 | NCBI Gene Expression Omnibus, GSM5972492 |
| Richard S | 2023 | Day 6 DMSO | https://www.ncbi.nlm.nih.gov/geo/query/acc.cgi?acc=GSM5972493 | NCBI Gene Expression Omnibus, GSM5972493 |
| Richard S | 2023 | Day 6 MS023 removed at day 4 | https://www.ncbi.nlm.nih.gov/geo/query/acc.cgi?acc=GSM5972494 | NCBI Gene Expression Omnibus, GSM5972494 |

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
