## [Editor Report · eLife assessment]

This **valuable** paper informs on the role of type I PRMTs in programming muscle stem cell identification. The evidence presented is mostly **solid**, with some weaknesses in the evidence regarding the proposed mechanism. The paper will be of particular interest to those who study skeletal muscle satellite cell biology.

---

## [Referee Report · Reviewer #1 (Public Review)]

In this study, Dominici et. al. show that small molecule inhibition of Type I PRMTs in muscle stem cells (MSCs) can result in the expansion of this cell type in vitro, solving a major limitation in the field. Importantly, once the inhibitor is removed these stem cells differentiate "normally". This advance will likely facilitate CRISPR-based screening approaches and stem cell engraftment therapy. Furthermore, they show that when a mouse model of Duchenne muscular dystrophy is treated with these same inhibitors these mice rather rapidly gain grip strength, demonstrating the therapeutic value of these findings.

Strengths:

- Previous studies from the same group have shown that the conditional ablation of PRMT1 in MSCs results in the expansion of this cell type, but this expanded PRMT1-null MSC pool cannot terminate the myogenic differentiation program. This raises the question of whether PRMT1 small molecule inhibition of MSCs will also facilitate the expansion of these cells, and if the removal of the inhibitor after expansion will result in a large functional pool of MSCs, which could then be used for both in vitro and in vivo studies.

- Using a combination of muscle fiber culture, myoblast culture, and single-cell RNA-seq, this is indeed what they show.

- They also perform two types of in vivo experiments to validate their cell culture findings; (1) MSCs expanded under the treatment of MS023 were washed clean of the inhibitor and engrafted into the tibialis anterior muscle. These cells were marked with GFP to allow efficient tracking. Mice receiving the MS023-treated MSCs produced more than double the mature GFP+ muscle fibers than cells treated with DMSO. (2) A mouse model of Duchenne muscular dystrophy displayed grip strength improvement after just one treatment of MS023.

- MS023 is a Type I PRMT inhibitor and thus can also target CARM1. CARM1 has been implicated in MSC function by the Rudnicki group. Importantly, they exclude a role for CARM1 in the expansion of MSC cell numbers by treatment with a very specific CARM1 inhibitor, TP064. Thus, indicating that PRMT1 inhibition is likely the main driver of this expansion phenotype.

---

## [Referee Report · Reviewer #2 (Public Review)]

In this manuscript, Dominici et al. aim to determine whether the reversible inhibition of the type I protein arginine methyltransferases (PRMT) would maintain the stemness of muscle stem cells in culture and enable subsequent regenerative capacities. They demonstrate that the type I PRMT inhibitor MS023 enhances self-renewal and in vitro expansion of muscle stem cells isolated from mice. Using a very rigorous single-cell RNA-sequencing approach, they further demonstrate that distinct sub-populations of cells emerge under type I PRMT inhibition and that these cells entered the differentiation program more efficiently. Moreover, they revealed a shift in metabolism in these cells, which they confirmed in vitro. Finally, they demonstrate that MS023 enhances muscle stem cell engraftment in vivo and that the direct injection of MS023 increases muscle strength in a mice model of Duchenne muscular dystrophy.

This study will have a great impact on the field of stem cells and offer potential therapeutic avenues for diseases such as Duchenne muscular dystrophy.

---

## [Referee Report · Reviewer #3 (Public Review)]

Dominici et al studied the effects of the type I PRMT inhibitor MS023 on skeletal muscle stem cells (MuSCs) and on muscle strength in dystrophin-deficient mdx mice. The authors observed an enhanced proliferative capacity of cultured MuSCs with an increase of Pax7+/MyoD- cells. The observations are more or less in line with previous studies of the same group, describing reduced differentiation but enhanced proliferation of MuSCs after genetic inactivation of Prmt1. scRNA-seq identified different subpopulations of MuSCs, showing a shift to increased Pax7 expression and elevated oxidative phosphorylation and glycolysis after treatment with MS023. Treatment of MuSC with MS023 during expansion in vitro enhanced engraftment of MuSCs and treatment of dystrophic mdx mice increased muscle strength.

Overall, the manuscript provides new insights into the beneficial effects of the type I PRMT inhibitor MS023 for skeletal muscle regeneration. The description of the MS023-induced transcriptional and metabolic changes in MuSC is interesting and the effects on MuSC transplantation and muscle strength are stunning. However, I have the following comments and concerns:

* Control experiments with the TP-064 inhibitor (previously shown to be specific for CARM1/PRMT4) were not done for the transplantation and muscle strength experiments, which is a clear shortcoming in my view. Since MS023 is a non-selective inhibitor of type I PRMTs with comparable IC50 values for PRMT1 and PRMT4 (CARM1), and lower IC50 values for PRMT6 and PRMT8, it is still not clear whether the enhanced transplantation efficiency and the increased muscle strength is indeed only caused by inhibition of PRMT1. The authors justify their statements by pointing out that gene expression of Prmt1 is highest among the type I PRMTs in MuSCs, which is a rather poor argument, as seen by the strong effects caused by the inactivation of PRMT4.

* Clustering of the M1-M5 subpopulations. I expressed my concern about the separation of the subclusters, which appear more or less in the same cloud. The authors answered that each cluster has some genes, which are only expressed in the respective cluster. I do not doubt this observation but apparently, the transcriptional differences are minor, otherwise one would have seen a much better separation of the subpopulations.

* The authors have not done additional experiments but simply toned-down the statements about the relevance of the proposed "metabolic reprogramming" of MuSC by the type I PRMT inhibitor MS023, which was a major conclusion in the original submission. Again, the changes in the expression of metabolically relevant genes upon MS023 treatment are interesting and should be analyzed in respect to causality. It is not a solution to more or less disabandon the original hypothesis by changing the wording.

* I specifically asked the authors to check whether the dramatic six-fold increase of MuSC engraftment after MS023 treatment really goes along with the incorporation of transplanted MuSC into the MuSC niche, raising concerns that a huge share of the transplanted cells may linger around in the interstitium. It should be very easy to identify and quantify transplanted MuSC outside and inside the basal lamina. Instead of doing the requested experiment, the authors argue about suppression of endogenous MuSC competition by irradiation, at the same time admitting that several GFP-negative fibers have formed.

* I expressed my doubts that a 3-day treatment with MS023 is sufficient to dramatically enhance muscle function in mdx mice via "improvement" of the MuSC population, as reported by the authors, even 30 days after administration of MS023. It seems much more likely that MS023 exerts additional effects that are responsible for the dramatic improvement of muscle function in mdx mice. I maintain my view that this needs to be interrogated more carefully since the improvement of muscle function of dystrophic mice is a central point of the study. It has to be made clear whether this is really due to "improved" functions of MuSC. Many other processes might be involved or responsible for the effect (e.g. impact on inflammation?).

---

## [Author Response]

The following is the authors' response to the original reviews.

We’d like to thank the three reviewers for reviewing in depth our work and providing insightful comments and suggestions.

**Reviewer #1 (Recommendations For The Authors):**
1. The evidence that MS023 is actually working in vivo in their last experiment (Fig 6) needs to be strengthened. This could be due to the timing of the experiment. Tail tips were collected 48 h after the final injection and analyzed by Western for ADMA and SDMA levels. They do see subtle changes, in the right directions, of SDMA and ADMA (but these changes are really not very obvious). Perhaps the inhibitor has already been largely metabolized two days after injection. Have they looked at MMA levels?

We have quantified the ADMA and SDMA levels of Fig. S6. We have not measured MMA levels. The text has been edited as follows:

“The average ADMA relative expression was 0.95 for vehicle treated mice and 0.83 for MS023 treated mice (p < 0.00041). The average SDMA relative expression was 0.92 for vehicle treated mice and 0.94 for MS023 treated mice (p < 0.17). These whole-body measurements as measured by tail biopsies show MS023 promotes the decrease of proteins with ADMA and a slight increase in proteins with SDMA. It is known that inhibition of type I PRMTs or PRMT1 deletion diminishes ADMA and increases SDMA due to substrate scavenging (Dhar et al, 2013).”2. The authors need to explain why they would expect an increase in SDMA levels in these mice after MS023-treatment.

We have edited the text as follows:

“It is known that inhibition of type I PRMTs or PRMT1 deletion diminishes ADMA and increases SDMA due to substrate scavenging (Dhar et al, 2013).”

3. In the discussion, it would be valuable to address the types of CRISPR-screens that could be performed in these MS023-expanded MSCs. They mention this as a benefit in the introduction, but to expand on this idea in the discussion.

The idea here was not necessarily to perform a CRISPR screen on the MS023-treated cells (although it is an interesting idea), but rather to correct the genetic mutation by CRISPR-Cas9 to enhance the success of genetically corrected autologous cell transplantation. The addition of MS023 to MuSC in vitro would allow to expand the cells while maintaining their self-renewal potential, thereby providing the opportunity to correct the mutation on the dystrophin gene using technologies such as CRISPR prime editing (Mbakam et al., 2022 Mol Ther Nucleic Acids 30:272-285). Our results demonstrating that MS023 enhances cell engraftment suggest that this method could be used to improve autologous cell transplantation efficiency. We have edited the text in the discussion as follows:

“Our findings suggest that type I PRMT inhibitors may have therapeutic potential for treating certain skeletal muscle diseases. For instance, to improve the efficacy of autologous cell therapy, the dystrophin-deficient MuSCs collected from DMD patient and corrected by CRISPR prime editing (Happi Mbakam et al, 2022) could be treated with MS023 to maintain their stemness and enhance their cell engraftment capacity.”

4. Also, could they address the potential value of MSC culture and expansion using a combination of SETD7 inhibition and PRMT1 inhibition?

Agreed. We have edited the text as follows:

“These findings suggest that inhibiting methyltransferases can affect MuSC fate and perhaps a combination of Setd7 and MS023 inhibitors would provide a more favorable combination to promote the expansion of MuSCs while maintaining their stem cell-like properties.”

**Reviewer #2 (Recommendations For The Authors):**
In figure 2 the authors show that upon removal of MS023, the cells differentiate more efficiently. In figure 5E-F they show that the mice that received MS023-treated cells had more GFP mature muscle fibers. However, in figure 5C-D these cells have the same capacities to terminally differentiate. This reviewer was wondering if these cells would differentiate faster? Have the authors look into this?

The reviewer raises an interesting point. Our in vitro experiments shown in Supplemental Figure S1 indicate that MS023-treated cells are actively more cycling (more ki67+ cells) and are less committed to differentiation (less Pax7-MyoD+ cells), which would suggest that they would need to exit the cell cycle and differentiate faster to reach the same fusion capacity after 3 days of differentiation without MS023. Future experiments with a time course including earlier time points will be needed to confirm if these cells differentiate faster.

**Reviewer #3 (Recommendations For The Authors):**
1. MS023 is a non-selective inhibitor of type I PRMTs. It has comparable IC50 values for PRMT1 and PRMT4 (CARM1), and lower IC50 values for PRMT6 and PRMT8. The authors argue that the cellular phenotype caused by MS023 is solely mediated via PRMT1, since the specific PRMT4-inhibitor TP-064 has no effects on MuSC expansion. TP-064 treatment was not used as a control for the transplantation and muscle strength measurement experiments. Are PRMT6 and PRMT8 expressed in MuSC and are thy inhibited by the applied concentrations of MS023? Kawabe et al reported that CARM1 methylates Pax7, thereby inducing Myf5 transcription during the asymmetric division of MuSC (PMID: 22863532). Is the expression of Myf5 reduced upon MS023 treatment? scRNAseq of MuSC 4-day after culture is too late to address this question, since the majority of the cells are already committed to differentiation. Staining for Myf5 using ex vivo cultured fibers or regenerating muscles in vivo should be used.

Indeed, we mention throughout the text that MS023 is a type I PRMT inhibitor. We have edited the text as follows suggesting the effect are most likely mediated by inhibition of PRMT1 *in vivo.*

“Treatment of MuSCs with MS023 resulted in metabolic reprogramming of MuSCs, supporting a role for type I PRMTs as metabolic regulators. In vitro, MS023 has been shown to inhibit several type I enzymes at nM concentrations (Eram et al., 2016). It is well-documented that PRMT1 is the major cellular type I enzyme (Pawlak et al, 2000) and this is why PRMT1, but not the other type I PRMTs are embryonic lethal in mice (Guccione & Richard, 2019). The numerous published data in cellulo with MS023 are thus far only reproduced by PRMT1-deficiency by siRNA or knockout, suggesting that MS023 actions in vivo are predominantly mediated by inhibiting PRMT1 (Gao et al, 2019; Plotnikov et al, 2020; Wu et al, 2022; Zhu et al, 2019). Thus, the effects of MS023 on MuSCs are most likely mediated by inhibition of PRMT1.”

Moreover, we investigated the expression of other type I PRMTs as suggested by the reviewer. We investigated their expression from publicly available single cell RNAseq dataset (Oprescu SN et al, iScience 2020, 23:100993), which performed analysis on skeletal muscle at different time points post-cardiotoxin injury (uninjured, and 0.5, 2, 3.5, 5, 10, 21 days post-injury). The findings show that *Prmt1* is by far the most expressed type I PRMT in MuSCs at every time point tested. *Carm1* (*Prmt4*) is expressed at high level in a small/moderate subset of cells, especially during regeneration. *Prmt6* is expressed at low level in a small proportion of cells, while *Prmt8* expression was not detected. These findings are coherent with our observation that Prmt1 is the predominant type I Prmt in MuSCs, which further support our hypothesis that it is the main target of MS023. These findings were added in Suppl. Fig 1B.

The expression of Myf5 during asymmetric division is indeed well characterized on muscle fiber-associated MuSCs (Dumont et al., 2015 Nat Med 21:1455; Kawabe et al., 2012 Cell Stem Cell 11:333). As the reviewer states, the 4-day time point is too late to investigate Myf5 expression. Additionally, these cells were cultured ex-vivo and were not fiber-associated. Therefore, scRNAseq is not an ideal method to address the question of whether MS023 treatment modulates Myf5 expression, and further experiments would be required to examine Myf5 in an appropriate context (i.e. on ex-vivo cultured myofibers).

2. Figure 2 is not very informative, while the second paragraph of the result parts is excessive and too complicated. The extensive description of differential gene expression in each potential subpopulation is neither very informative nor helpful to convince the reader that the M3/M5 population has acquired more stemness-like features due to the MS023 treatment. From my point of view, the data just reflect the increased proliferative capacity of MS023-treated cells with elevated cell cycle markers, ribosomal protein, and metabolic state. Do the M1-M5 populations show any different distribution along the trajectory? The authors need to show cell trajectories for each sample and cluster in Figure S3A. It is also imperative to present the distribution of signature genes for each individual cluster. Essentially, M1-M5 all located together in one cloud. What justifies segregation into different subclusters? The color code for the different clusters (including the trajectories) to allow better distinction.MS023 treated MuSCs contain a subpopulation with higher Pax7 expression (Supplementary Figure S2F, S2G), which is consistent with the IF results in Figure 1 and emphasized in the abstract. Why are these data in the supplements and not in a main figure (e.g. in figure 2)?

We appreciate the thoughtful and detailed comments on our single-cell data. Please see below for a response to each point:

To address the concern that the results section is excessive, our intention was to simply provide the reader with a descriptive overview of the identity of each subcluster that the software identified. In fact, to ensure clarity and conciseness, we elected to provide only the names of a select few cluster markers rather than list all of the significant cluster markers that were generated. We kindly refer the reviewer to Supplementary Table S1 for a more extensive list of markers.

In response to the reviewer’s comment: “The color code for the different clusters (including the trajectories) to allow better distinction,” we agree that colour-coding is helpful, please refer to Figure 2A for a colour-coded map of the clusters.

To address the reviewer’s question regarding what justifies segregation into different subclusters for M1-M5, refer to Supplementary Table S1 for a list of uniquely enriched markers for each cluster. This list was filtered to include marker genes that were present only in a given cluster, thus contributing to its uniqueness and explains why that cluster was identified as being distinct from another given cluster.

Lastly, since the elevated Pax7 levels in MS023-treated MuSCs was already presented and discussed thoroughly in Figure 1, we elected to avoid repetition in the main Figures and presented the ridge plots showing elevated Pax7 in the Supplementary Material for Figure 2

3. The same group has reported previously that PRMT1-deficient MSCs show reduced expression of MyoD due to disruption of Eya1/Six1 recruitment to the MyoD promoter (PMID: 27849571). However, the scRNAseq result does reflect this finding. MyoD levels are not significantly changed in d4 MS023 compared with d4 (Supplementary figure S2G). The authors need to provide an explanation. Furthermore, the authors previously described that "the majority of PRMT1-deficient MSCs repressed Pax7 expression at day 3 while being Ki67 positive (Fig. 5B). How does that fit to the current observations, which indicate an increase of Pax7+ cells after MS023 treatment? This discrepancy needs to be resolved.

While the scRNAseq does not show a reduction in overall MyoD expression in MS023-treated MUSCs, there is indeed a reduction in the proportion of MyoD+ myofiber-associated MuSCs (Figure 1C, 1D). Supplemental Figure S2G further shows a subpopulation in the d4MS023 group with lower MyoD expression that was not present in the d4 group, reflective of the findings in Figures 1C and 1D. Therefore, although the average expression was not shifted significantly with MS023, there was indeed a subpopulation of MuSCs with lower MyoD expression.

The reviewer additionally points out that Fig. 5B from a previous study (Blanc et al., 2017 MCB 37:e00457) performed by our group, shows that Pax7 expression was repressed at day 3 of culture in PRMT1-null MuSCs. However, this quantification was based on immunofluorescence staining where cells are marked positive or negative for Pax7 expression and does not look at the intensity of Pax7 expression levels. In our current study, we examine the expression levels of Pax7 in discrete subpopulations of MuSCs and found that there is a subpopulation of MuSCs that emerges with MS023 treatment that has higher Pax7 expression than untreated counterparts. Therefore, the results of the two experiments are not directly comparable.

4. I do have a major problem with the interpretation of the metabolic changes in MS023-treated MuSC. In the abstract, the authors wrote, "These findings suggest that type I PRMT inhibition metabolically reprograms MuSCs resulting in improved self-renewal and muscle regeneration fitness." There is simply no causal evidence to support this claim, which is solely based on a correlation. If the authors want to maintain this claim they either need to stimulate OXPHOS and glycolysis by other means to see whether such a manipulation recapitulates the effects of MS023 or attenuate OXPHOS and glycolysis to see whether this abrogates the effects of MS023. To prove whether increased OXPHIS is a cause for improved self-renewal, the authors might simply co-treat MuSC with MS023 and an OXPHIS inhibitor and analyze consequences for the Pax7+/MyoD- population.

We thank the reviewer for the excellent suggestions of experiments that would solidify a causal relationship between increased metabolism and increased self-renewal. We will certainly consider them for future studies. We agree that the relationship in the present study is correlative, and the text has been modified in the abstract as follows:

“Single cell RNA sequencing (scRNAseq) of ex vivo cultured MuSCs revealed the emergence of subpopulations in MS023-treated cells which are defined by elevated Pax7 expression and markers of MuSC quiescence, both features of enhanced self-renewal. Furthermore, the scRNAseq identified MS023-specific subpopulations to be metabolically altered with upregulated glycolysis and oxidative phosphorylation (OxPhos). Transplantation of MuSCs treated with MS023 had a better ability to repopulate the MuSC niche and contributed efficiently to muscle regeneration following injury. Interestingly, the preclinical mouse model of Duchenne muscular dystrophy had increased bilateral grip strength 10 days after a single intraperitoneal dose of MS023. Our findings show that inhibition of type I PRMTs increased the proliferation capabilities of MuSCs with altered cellular metabolism, while maintaining their stem-like properties such as self-renewal and engraftment potential.”

5. Ryall et al reported that MuSCs undergo a metabolic switch from fatty acid oxidation to glycolysis with reduced intracellular NAD+ levels and reduced activity of SIRT1, leading to elevated H4K16 acetylation. Here, both OXPHOS and glycolysis are increased after treatment of MuSC with MS023. Are the NAD+ and H4K16ac levels changed in MS023-treated MuSC?

This is another excellent study that would help to support a causal relationship between MS023 treatment and increase OXPHOS and glycolysis and could certainly be addressed in future studies.

6. In Ryall et al.'s results, there was no difference in the basal mitochondrial OCR between freshly isolated MuSCs and cultured MuSCs. Importantly, stimulation of OXPHOS will increase ROS concentration, resulting in premature differentiation of MuSC (PMID: 30106373). Furthermore, increased ROS levels will most likely enhance DNA damage rather than improve self-renewal. The authors have to address these issues and also monitor ROS and DNA damage levels.

The lack of cell death upon treatment with MS023 in the present study would indicate that there is no major ROS-induced DNA damage occurring. Additionally, the propensity of MS023-treated MuSCs to retain their stemness while in long-term culture (Supplemental figure S1E) would indicate that in this context, premature differentiation is not a concern.

7. The authors used FACS-analysis of MuSCs three weeks after transplantation to demonstrate that MS023 treatment enables better engraftment into the MuSC niche. The six-fold increase of transplanted cells in the MuSC niche is difficult to understand, Why shall transplanted cells compete so efficiently with endogenous MuSC for repopulation of the niche? Is it possible that some of the transplanted MuSC are still lingering within the interstitium and erroneously counted as bona fide MuSC? The authors have to determine the localization of transplanted MuSC. Are all transplanted cells indeed situated in the proper niche or are they also present outside the basal lamina of muscle fibers?

The hindlimbs which received the engraftment were irradiated 24h prior to engraftment, therefore the ability of endogenous MuSCs to compete is compromised. Additionally, Figure 5E shows that the regenerated muscle indeed has GFP negative fibers that would have been generated from endogenous MuSCs, indicating that MS023-treated MuSCs did not fully outcompete endogenous MuSCs.

8. The authors reported that an only 3-day treatment with MS023 is sufficient to dramatically improve muscle function in mdx mice even 30 days later, which is hard to swallow. What is the evidence that such strong effects are primarily mediated by stimulation of MuSC expansion? Are there other pathways or cells that respond to MS023 treatment and stimulate muscle strength? To support the claim of a 'better' stem cell function as the major cause for MS023-dependent stimulation of muscle strength in mdx mice, the authors need to determine the total number of Pax7+ cells, Pax7+/Ki67+, Pax7+/MyoD+, Pax7+/MyoD-, Pax7-/MyoD+ and myonuclei. It is also absolutely mandatory to include wildtype controls in the muscle strength measurements. Does MS023 treatment also increase muscle strength in wild-type controls?

Agreed. We cannot exclude if the effect is mediated by an expansion of the MuSC pool or by an effect on other cell types, such as a direct impact on the myofibers. The manuscript has been modified to include the following text:

“Furthermore, our findings show that injection of MS023 in the dystrophic mouse model mdx led to enhanced muscle strength with effects lasting up to 30 days. We cannot exclude if the effect of MS023 was mediated by an expansion of the MuSC pool or by an effect on other cell types, such as a direct impact on the myofibers. The goal of this experiment was to provide a therapeutic perspective for the possible use of type I PRMT inhibitor for the treatment of DMD.”

The goal of this figure was to provide a therapeutic perspective for the use of type I PRMT inhibitor for the treatment of DMD. Muscle wasting/weakness in DMD is a complex and multifactorial process (e.g., myofiber fragility, MuSC defects, chronic inflammation, fibrofatty accumulation). If MS023 can target multiple aspects of the physiopathology of the disease it would increase its therapeutic applicability. Further studies will be needed to determine the exact mechanism by which MS023 mediate its beneficial effect. These future studies could include the use of wild type control, as the reviewer suggests, to investigate the role of MS023 in a non-muscle degenerative context.

9. Ideally, a genetic inactivation-reactivation of PRMT1 should be done to validate the results with MS023 and to make sure that indeed the transient inhibition of PRMT1 is responsible for the beneficial effects of MS023. Of course, this would be a major effort when done in genetically manipulated mice and therefore is not adequate to ask for. However, it should be possible to use PRMT1-deficient MuSC, which the authors have in hand, and re-express PRMT1 in these cells with an AAV or a lentivirus.

We agree that genetic ablation of PRMT1 is a key experiment to validate MS023 results. Please refer to previous work from our group, which shows that PRMT1-KO MuSCs have an enhanced self-renewal phenotype (Blanc et al., 2017 MCB 37:e00457), similar to what was observed in the present study with MS023 treatment.

10. Some claims are overstated and/or to aggressive. E.g.: "Therefore, through repression of type I PRMTs with MS023, we have reprogramed MuSCs to acquire a unique and previously uncharacterized identity." I do not see clear evidence that MS023 treatment 'reprograms' MuSC to a 'unique identity'. The observed changes are in large parts compatible with a simple stimulation of proliferation.

The unique finding in our data is that treatment with MS023 resulted in a shift in identity as compared to the DMSO-treated proliferating MuSCs (M1, M2 and M4), creating transcriptionally distinct M3 and M5 clusters. M3 and M5 had elevated markers for metabolism (E.g. Eno1, Atp5k, etc) and early activation (E.g. Fos, Jun), while the untreated MuSCs in clusters M1, M2 and M4 did not. Furthermore, M3 and M5 had higher baseline levels of Pax7 expression when compared to untreated cells. Together, these findings describe a transitional subpopulation of MuSCs unique to MS023 treatment which not only harbour stem like/early activation markers Pax7, Fos and Jun, but also elevated proliferative markers related to cell cycle and energy metabolism. This particular combination of characteristics is unique to the MS023-treated MuSCs, thus identifying a unique subtype of MuSC identity. In accordance with our scRNAseq data, we validated experimentally that MS023-treated cells have higher energy metabolism and increased self-renewal potential, thereby confirming that the unique transcriptomic signature of these cells also lead to a different cell fate decision.